# Enhancement of large-scale flood risk assessments using building-material-based vulnerability curves for an object-based approach in urban and rural areas

Johanna Englhardt[1], Hans de Moel[1], Charles K. Huyck[2], Marleen C. de Ruiter[1], Jeroen C. J. H. Aerts[1], Philip J. Ward[1]

[1]Institute for Environmental Studies (IVM), Vrije Universiteit Amsterdam, De Boelelaan 1087, 1081HV Amsterdam, The Netherlands
[2]ImageCat Inc., Long Beach, CA 90802, USA

*Correspondence to*: Johanna Englhardt (j.englhardt@vu.nl)

**Abstract.** In this study, we developed an enhanced approach for large-scale flood damage and risk assessments that uses characteristics of buildings and the built environment as object-based information to represent exposure and vulnerability to flooding. Most current large-scale assessments use an aggregated land-use category to represent the exposure, treating all exposed elements the same. For large areas where previously only coarse information existed such as in Africa, more detailed exposure data are becoming available. For our approach, a direct relation between the construction type and building material of the exposed elements is used to develop vulnerability curves. We further present a method to differentiate flood risk in urban and rural areas based on characteristics of the built environment. We applied the model to Ethiopia, and found that rural flood risk accounts for about 22% of simulated damage; rural damage is generally neglected in the typical land-use-based damage models particularly at this scale. Our approach is particularly interesting for studies in areas where there is a large variation in construction types in the building stock, such as developing countries.

## 1. Introduction

Globally, floods are one of the main natural hazards in terms of socioeconomic impacts, causing billions of dollars of damage each year. For example, between 1980 and 2013, global flood damage exceeded $1 trillion, and resulted in ca. 220,000 fatalities (Dottori et al., 2016). Reducing disaster risk, such as from flooding, is globally very high on the political agenda. For example, it is an important aspect of both the Sendai Framework for Disaster Risk Reduction (UNISDR, 2015) and the Warsaw International Mechanism for Loss and Damage Associated with Climate Change Impacts (UNFCCC, 2013). To achieve this reduction in risk at the global scale requires methods to quantitatively assess global flood risk (Mechler et al., 2014). Here, flood risk is defined as a function of three components: hazard (e.g. flood extent and depth), exposure (assets and people exposed), and vulnerability (factors that determine the susceptibility of the exposed assets to the hazard) (UNISDR, 2015).

Global flood risk assessments are increasingly used in decision-making and practice, and have been useful for identifying flood risk hotspots (e.g. Ward et al., 2015). In an ideal situation, such flood risk assessment models could use detailed, high-resolution data for all locations around the globe (Jonkman, 2013). In practice, data and resources required for such models rarely exist, and therefore global flood risk models have been developed. Current global flood risk models often use

resolutions between 30" x 30" and 0.5° x 0.5° to assess the exposed assets (e.g. Alfieri et al., 2013; Arnell and Gosling, 2016; Ward et al., 2013). Recently, much effort has been put into improving global risk models, mainly by improving the hazard component (e.g. Dottori et al., 2016; Ikeuchi et al., 2017; Sampson et al., 2015; e.g. Trigg et al., 2016). However, much less attention has been given to improvements in the representation of exposure and vulnerability, despite the fact that their overall contribution to uncertainty is large (de Moel and Aerts, 2010).

In large-scale assessments, i.e. regional to global levels, exposure is generally represented based on aggregated land-use categories, especially in regions where only limited data are available, such as Africa (de Moel et al., 2015). Whilst using such data provides a useful first assessment of large-scale damage and risk (e.g. Feyen et al., 2011; Hall et al., 2005; Ward et al., 2013), more detailed information of the exposed objects could improve these assessments. Vulnerability is mostly represented using stage-damage functions, also known as vulnerability curves, which describe the relationship between the

potential damage of the exposed elements for different levels of the hazard (usually water depth). These functions can represent physical vulnerability, which we refer to in this paper, however not social vulnerability (i.e. characteristics that influence a person's or group's capability of dealing with the impact of a natural hazard), or other vulnerability dimensions (e.g. institutional, economic, environmental) (Fuchs, 2009; Papathoma-Köhle et al., 2017). For large-scale studies, a vulnerability curve is generally developed for each of the aggregated land-use categories used to represent exposure (Ward et

al., 2013).

Whilst aggregated land-use categories may be a suitable option to represent exposure if data are limited, they cannot reflect the (spatial) heterogeneity within each land-use category (Wünsch et al., 2009). For instance, large-scale flood risk models usually focus on an 'urban' category that aggregates exposed elements of various types (e.g. houses, infrastructure, shops, green areas etc.) into one land-use class (Ward et al., 2015). Since an aggregated land-use category like 'urban' is coupled to

one 'urban' vulnerability curve, these curves generalise the relationship between flood depth and damage across all of the diverse exposed element types within that category. Without a more direct relation between these types of exposed elements and the impact of flood waters, large uncertainties exist in the simulated damage (de Moel and Aerts, 2010). More detailed information on the specific land use, its extent, and the vulnerability of the exposed elements could improve large-scale assessments, for example by using high-resolution remote sensing products (Goldblatt et al., 2018; Myint et al., 2011) or

information as used in local-scale flood damage studies at an object level (individual buildings, businesses, infrastructure objects, etc.) (de Moel et al., 2015; Merz et al., 2010). In our approach, we therefore utilize information about the composition of an area's building stock and the characteristics of exposed objects, particularly construction types and materials. Applying such object-based information, which is not to be confused with object based image analysis in remote sensing, is contrasting to the common land-use-based approach in large-scale flood risk assessments.

The literature distinguishes flood vulnerability of buildings according to different structural factors (such as building type, quality, height, and material), as well as occupancy type (such as residential, commercial, industrial, etc.). The latter is a commonly used factor for determining vulnerability (de Ruiter et al., 2017), with much fewer studies relating potential losses to the structural factors. Reasons for this are the paucity of information and the huge effort it takes to obtain information on

the damage incurred by individual objects and the structural components (Wahab and Tiong, 2016). Studies or models that do include information on these factors are mostly based on surveys and have therefore only been feasible on smaller scales. FLEMOps (Thieken et al., 2008) is an example of a model that uses survey data on flood damage in Germany, and includes factors such as building type and quality. The study by de Villiers et al. (2007) is one of the few assessments (see also World Bank, 2000) within Africa, but uses size and content value of houses to determine flood damage and does not go into detail

on structural features. Studies that focus on construction type and building material to assess the flood damage show that these characteristics, together with ground floor elevation and number of floors, are important features in determining the vulnerability of different building types to floods (e.g. Godfrey et al., 2015; Neubert et al., 2008; Schwarz and Maiwald, 2008; Zhai et al., 2005). Furthermore, building characteristics are essential components of physical vulnerability and risk assessment in the earthquake domain (de Ruiter et al., 2017), as well as for other flood types such as flash floods in mountain

areas and debris flows. For such studies on the local-scale aspects can even include for example features of the building envelope such as layout of openings and wall dimensions, flow direction, sediment load and surrounding buildings; these elements are sometimes evaluated via laboratory experiments and on-site data collection (e.g. Godfrey et al., 2015; Milanesi et al., 2018; Sturm et al., 2018). There is a gap in applying such indicators in large-scale flood risk assessments, which could be improved by using object-based characteristics to represent exposure and vulnerability, particularly in developing

countries with a diverse structural building stock.

Recently, a building exposure dataset has been developed for several African countries as part of the Building Disaster Resilience program for the World Bank's Africa Disaster Risk Financing Initiative by ImageCat (ImageCat et al., 2017). ImageCat uses a stratified sampling technique that infers the number of buildings in a region from census data and then uses image processing tools to identify development patterns (Hu et al., 2014). The construction practices are then characterized

through a review of the literature, interviews, review of VHR images, in situ video, and in some cases site visits (Silva et al., 2018). This characterization of development patterns is used for dasymetric mapping of building counts to a 15" grid. Estimates are supplemented with total estimates of floor area, and replacement values based on construction practices observed in each development pattern (Huyck and Eguchi, 2017). Compared to the methods employed in current large-scale flood risk models, the information about the built environment of an area and its characteristics as provided in such datasets

enables a differentiation between the exposed objects in terms of vulnerability to flood waters and exposed value.

Furthermore, a greater level of detail opens up the possibility to address the issue of distinguishing urban and rural flood risk. This is commonly neglected in land-use-based flood risk assessment, due to the focus on higher value urban damage. Moreover, land-use classification studies have difficulties in assessing urban and rural differences. This is because the resolution in previous land-use and land-cover products was not sufficient to identify smaller settlements, and the

characteristics of urban and rural areas are very different and can be difficult to grasp in land-use classification studies (Dijkstra and Poelman, 2014). Internationally there is no agreed way to distinguish urban from rural areas. For example, according to the national census of Ethiopia, localities of 2,000 or more inhabitants are considered urban, whereas the urban definition for Niger only includes capitals of departments and districts (UNSD, 2016). Another traditional distinction is that

urban areas provide a different way of life and usually a higher living standard (UNSD, 2017). Compared to developed countries, the building stock in rural areas of developing countries is often constructed from more traditional and less expensive building materials, which makes them more vulnerable to flooding. In this regard, urban settlements in the context of this study are defined as geographic units with built-up area that are more developed and have a higher built-up density than rural settlements.

The aim of this paper is to develop an approach for assessing large-scale river flood risk in urban and rural areas using object-based data from ImageCat to represent exposure, and to develop vulnerability curves for different building classes. The approach draws upon common practices in earthquake risk assessments, and relates damage by flood waters more directly to the vulnerability of buildings based on the building materials. We test the suitability of this approach for the case of Ethiopia, comparing our results with those using a more traditional large-scale flood risk modelling approach, examining

how the increased detail influences risk estimates. In addition to river floods, Ethiopia has experienced flash flood events in the past such as in 2006 with several casualties and millions of property damage in Dire Dawa (Billi et al., 2015). However, these kinds of floods are not included in this analysis.

## 2. Data and Methods

The approach used in this study is composed of the following main four steps, and shown in Figure 1:

1) development of **vulnerability classes and curves** for different construction types and building materials based on a literature review of previous studies;

2) classification of an **object-based exposure** dataset using input data from ImageCat;

3) derivation of **maximum damage values** and

4) **risk assessment** by combining the aforementioned vulnerability and exposure with hazard data.

Each of these steps is described in more detail in the following subsections.

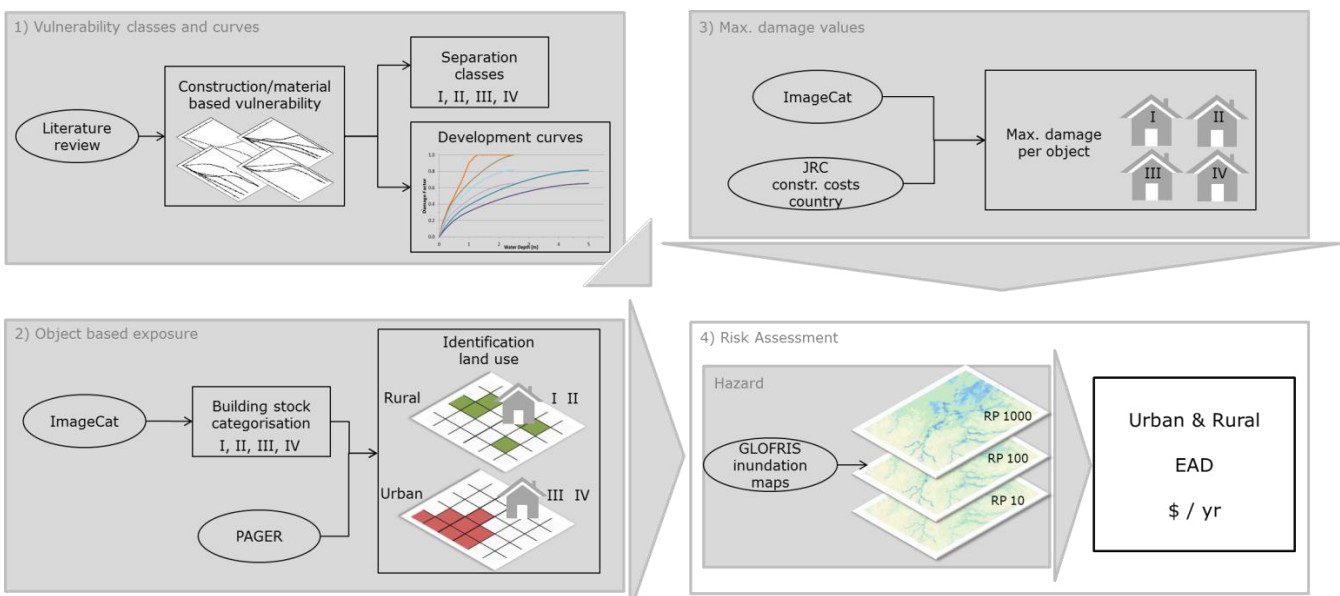

**Figure 1 Flowchart for large-scale flood risk assessment using object-based data with a building-material-based vulnerability approach.**

## 2.1. Vulnerability classes and curves

5     As a first step (Figure 1), an extensive literature review was conducted to develop flood vulnerability classes and associated curves based on construction types and building materials (Table 1). An increasing number of studies investigate multi-parameter damage models (e.g. Chinh et al., 2016; Wagenaar et al., 2018), but given the large amount of data required to apply such models, we here only consider studies on river floods that apply stage-damage curves. For the class and curve development, we use studies from different regions that have focused on the vulnerability of individual construction types or

10   building materials, and which are preferably based on actual event data. Some additional studies, often more qualitative in nature, were used to further refine the flood vulnerability classifications of the different building materials and construction types (e.g. Kappes et al., 2012; Laudan et al., 2017; Neubert et al., 2008; Zhai et al., 2005). Apart from reviewing the literature, experts with a structural engineering background were consulted to confirm the coherence of the final classification and vulnerability curves.

**Table 1 Overview of studies used to derive construction type and building-material-based vulnerability classes and curves. The four classes are: (I) non-engineered buildings created by compacted mud, adobe blocks or informal buildings; (II) wooden buildings; (III) unreinforced masonry/concrete buildings; and (IV) reinforced masonry/concrete and steel buildings**

| Vuln. class | Country | Source | Data basis | Main structural type / bldg. material | Event / applied area |
|---|---|---|---|---|---|
| I | India | Dhillon (2008) | Field survey | Mud structures | Birupa River basin in Orissa after the 2006 flood |
| I | India | Maiti (2007) | Household interviews | Mud wall buildings | Rural areas in Orissa after the 2003 flood |
| I | China | Li et al. (2016) | Interviews, questionnaires, field investigation | Wood-earth structures | Taining county town, Fujian province |
| I | Malawi | Rudari et al. (2016) | To generic Malawi housing typology adjusted CAPRA | Traditional (mud walls), semi-permanent (sun-dried bricks) typologies | Based on data for Northern and Central Malawi |
| II | India | Dhillon (2008) | Field survey | Wooden structures | Birupa River basin in Orissa after the 2006 flood |
| II | Germany | Buck (2007) | Expert seminar | Wood structures | Bldgs. in flood prone areas of Greifswald |
| II | New Zealand | Reese and Ramsay (2010) | Based on international studies and adjusted by post-event surveys | Timber buildings | Hutt Valley flood risk case study using major flood events in 2004 and 2007 |
| II | Australia | Hasanzadeh Nafari et al. (2016) | Derived data of extreme events and other models | Timber wall structures | Queensland 2013 |
| II | Japan | Dutta et al. (2003) | Function derived from post flood event data | Wooden structures | Applied to case study area in Chiba prefecture |
| II | Guatemala | Peters Guarín et al. (2005) | Field survey, interviews | Wood frame and board construction | Flood in Samalá River tributaries related to precipitation of hurricane Mitch 1998 |
| II | Philippines | Sagala (2006) | Field survey, household interviews | Wood, bamboo structures | Floods in 1995 and 2004 at Naga and Bicol River in Sabang and Igualdad Barangay, Naga City |
| II | Romania | Godfrey et al. (2015) | Expert weighted vuln. index and curves from other studies | Wooden buildings | Applied to case study in Nehoiu Valley |
| III | India | Dhillon (2008) | Field survey | Brick, cement structures | Birupa River basin in Orissa after the 2006 flood |
| III | Australia | Hasanzadeh Nafari et al. (2016) | Derived data of extreme events and other models | Masonry buildings | Queensland 2013 |
| III | Bangladesh | Islam (1997) | Household and expert interviews | Brick buildings | Floods between 1988 and 1993 in urban areas |
| III | China | Li et al. (2016) | Interviews, questionnaires, field investigation | Brick-wood and masonry structures | 2010 flood in Taining county town, Fujian province |
| III | Australia | Middelmann-Fernandes (2010) | Based on quantity surveyor data | Brick-veneer structures | Swan River system in Perth, Western Australia |
| III | Malawi | Rudari et al. (2016) | To generic Malawi housing typology adjusted CAPRA | Permanent (burnt bricks, concrete, stone walls) typologies | Based on data for Northern and Central Malawi |
| III | Philippines | Sagala (2006) | Field survey, household interviews | Concrete structures | Floods in 1995 and 2004 at Naga and Bicol River in Sabang and Igualdad Barangay, Naga City |
| IV | China | Li et al. (2016) | Interviews, questionnaires, field investigation | Steel-reinforced concrete structures | 2010 flood in Taining county town, Fujian province |
| IV | India | Maiti (2007) | Household interviews | RCC structures | Rural areas in Orissa after the 2003 flood |
| IV | Germany | Buck (2007) | Expert seminar | Reinforced masonry / concrete structures | Bldgs. in flood prone areas of Greifswald |
| IV | Japan | Dutta et al. (2003) | Function derived from post flood event data | RC concrete buildings | Applied to case study area in Chiba prefecture |

Table 1 summarises the studies used to derive construction type and building-material-based vulnerability classes and curves. In all of these studies, the construction type or (dominant) building material is clearly specified, and is either the only indicator, or one of the primary indicators, for the description of the flood vulnerability. Four vulnerability classes can be identified from this literature, of which each class consists of similar construction types and building materials with comparable behaviour towards flooding. The four classes are: (I) non-engineered buildings built with materials such as compacted mud and adobe block or informal buildings; (II) wooden buildings; (III) unreinforced masonry/concrete buildings with walls of burnt bricks or stone or concrete blocks; and (IV) reinforced masonry/concrete and steel buildings.

From the literature described in Table 1, we identified information to develop the stage-damage curve for each of these vulnerability classes. The stage-damage curves in most of the studies are concave, increasing steeply at low water depths (especially for the buildings made with more vulnerable materials), and with a decreasing slope at higher water depths. This overall concave shape was differentiated into curves for each of the four vulnerability classes, shown in Figure 2, using information on threshold levels (e.g. the water depth at which most damage was incurred) from the studies in Table 1. We distinguish curves that go up to 2.5m and up to 5m (for buildings with 1- and 2-floors) as flood levels rarely reach higher levels. Housing built through informal channels dominate in Africa (World Bank, 2015), and self-constructed buildings using inexpensive materials and traditional manufacturing techniques are still very common (Alagbe and Opoko, 2013; Collier and Venables, 2015). Buildings of class I and II belong to this group and are assumed to be one floor only, as multiple story buildings would require higher quality materials and hiring a professional construction crew. The four vulnerability classes are described below:

*Class I* are non-engineered buildings built with materials such as compacted mud, (non-cemented) adobe blocks and other traditional materials found in the natural environment or informal buildings (often using natural or scrap materials for the walls and roof covers). Buildings in this class can disintegrate and collapse easily when impacted by flood waters and thus are the most vulnerable to flooding. Literature shows that mud walls can collapse when flooded by about a meter of water (Maiti, 2007), and submersion tests illustrate that most adobe bricks completely dissolve when submerged for 24 hours (Chen, 2009). Depending on the material mixture and mortar for example by adding cement the stability of these buildings can be increased. However, with the high level of the cement prices in Africa (Schmidt et al., 2012) this is rather consideration for class I buildings in other regions. Buildings of class I are assumed to be one floor only.

*Class II* consists of wooden buildings. Theoretically, these are far less vulnerable to collapsing than class I, when held together by joinery or nailing and straps into a structural frame and have durable wall and roof cover materials, but if wood frames become wet, they often have to be replaced, or finishing needs to be removed for drying (and replaced afterwards). In a study carried out in Germany, Buck (2007) showed that the damage can be ~35%-50% higher for wood frame homes than for masonry/concrete homes. However, the value and quality of the wooden buildings in Ethiopia is much lower and they seem to be predominantly present in rural areas with more informal, less durable building material. Therefore, we decided to let the curve progress up to damage factor 1 (total loss due to destruction or need for demolition) at flood depth of 2.5 m (i.e. damage can reach full building value, unlike masonry and concrete constructions). Buildings that are based on wood

construction types can account for a large proportion of overall building stock in some countries (e.g. USA, Japan and Ethiopia). The quality of these constructions and the building's value can vary considerably. For large-scale assessments outside of Africa, adjustment towards a greater flood resistance is recommended.

*Class III* are unreinforced masonry/concrete buildings with walls of burnt bricks or stone or concrete blocks. These buildings
are more vulnerable than those in class IV (reinforced masonry/concrete or steel). This is related to the fact that unreinforced walls are less able to resist the pressure of flood water exerted on walls. However, damage potential is assumed to be less than class II, as bricks, stone and concrete blocks are more durable and less likely to disintegrate or need replacement after being flooded compared to wood. Nonetheless, as described in Li et al. (2016), brick masonry buildings are less resilient than steel-reinforced structures. Therefore, a curve between class II and class IV was created for both one and two storey
buildings of this class.

*Class IV* represents engineered reinforced masonry/concrete and steel buildings. These types of buildings are engineered and basically standard in most western countries and large cities in Africa. Overall, they constitute the most resistant class to flooding. Many studies (e.g. Buck, 2007; Li et al., 2016; Maiti, 2007) show that vulnerability curves for these types of buildings do not go up to a damage factor of 1, as some elements do not need replacement after a flood (e.g. the foundation
or the structural walls or the frames). This is similar to the values from Dutta et al. (2003) and HAZUS-MH (Scawthorn et al., 2006), who show examples of curves that go up to 0.6-0.7 damage ratio. Therefore, in this study it is chosen to let this curve go up to 0.65. Both reinforced masonry and reinforced concrete and steel are put in the same class.

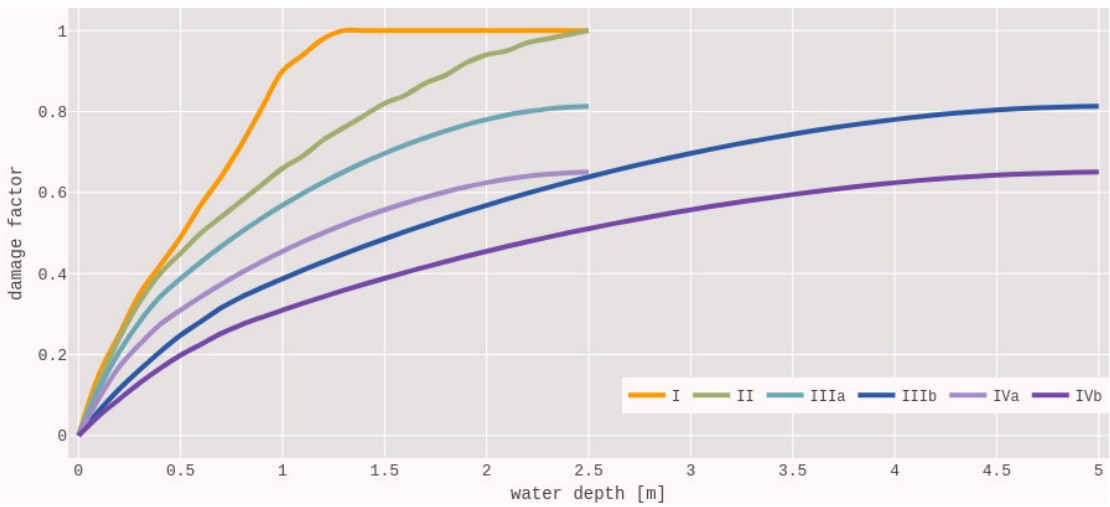

**Figure 2 Stage-damage curves for four building-material-based vulnerability classes. For class III and IV the one and two floor curve are denoted by (a) and (b).**

## 2.2. Object-based exposure data

In step 2 (Figure 1), we reclassify the objects identified in the ImageCat database into the four vulnerability classes, and distinguish between urban- and rural areas. The exposure data developed by ImageCat are available on a 15" x 15" grid for several African countries. Each grid cell contains building counts for different construction types, as well as the total floor area and total building value of the cell's building stock. For the building numbers the Ethiopian census data on housing units was used directly in most regions as the building stock is mostly single family housing. The living area was gleaned from sampling building footprint data, and as with structural characteristics varies by development pattern. For a predominantly commercial pattern, building stock data is adjusted with exposure derived from building footprint data. The number of floors can vary by development pattern, but for the vast number of buildings is single story for most of the country. For highly urbanized areas the number of stories was adjusted through expert opinion of several country-based structural engineers (Huyck and Eguchi, 2017). In total, 22 construction types are differentiated in the ImageCat data. Table 2 shows how these can be reclassified into the four vulnerability classes used in our study. Further description of the construction types can be found in supplementary section 1. In the Ethiopian data nine of the types from Table 2 occur.

**Table 2 Construction types of the ImageCat building exposure data with their respective flood vulnerability class.**

| Type | Description | Vuln. class | Type | Description | Vuln. class |
|------|-------------|-------------|------|-------------|-------------|
| ADB | URM adobe building | I | DS | Stone masonry building | III |
| ERTH | Earthen building | I | STN | URM stone building | III |
| INF | Informal building | I | UCB | Unreinforced concrete block building | III |
| M | Mud walls building | I | UFB | Unreinforced fired brick masonry building | III |
| RE | Rammed earth building | I | BTLR | Steel frame with bracing rods (Butler) building | IV |
| WWD | Wattle & daub building | I | C2 | Reinforced concrete shear wall building | IV |
| W2 | Wood frame building | II | C3 | Non-ductile RC frame with masonry infill walls building | IV |
| WLI | Light wood building | II | MCF | Confined masonry building | IV |
| WS | Solid wood building | II | RC | Reinforced concrete frame with URM infill building | IV |
| BRK | URM brick building | III | RM | Reinforced masonry brick building | IV |
| CB | URM concrete block building | III | S | Steel building | IV |

Most large-scale flood assessments focus on urban areas due to the availability of data and high potential damage. In countries with large differences between urban and rural living standards, such as developing countries, it can be expected

that the share of more vulnerable buildings (i.e. class I and II) is higher in rural areas compared to urban areas (e.g. Fiadzo, 2004). To account for these differences, we classify each cell as urban or rural. If more than 50% of the ImageCat objects in a cell belong to vulnerability class I or II, the area is assumed to be predominantly rural.

To check the assumption that the share of class I and II buildings in developing countries is higher in rural areas compared to urban areas, we examined these shares in the PAGER dataset (Jaiswal and Wald, 2008; Jaiswal et al., 2010). PAGER is a global residential and non-residential building inventory at the country level (usually but not exclusively expressed in proportions of people living or working in particular building structure typologies in urban and rural areas respectively), which is often used in earthquake research. PAGER provides information at a country level on the construction types that make up the total urban and rural building stock, though the information quality is varying between countries. First, we reclassified the PAGER construction types into the four flood vulnerability classes used in our study (see Supplementary table 1). Then, we calculated the percentage of buildings in PAGER's total urban and rural building stocks that are categorised as class I and II (Figure 3). The building stock differences between urban and rural areas can be found to a changing degree in all groups. While there is a distinct gap suggested for Africa, PAGER has to rely there on very limited information (i.e. only 2 of the countries differentiate urban and rural building stock without judging on information from neighbouring countries). Nevertheless, the data for urban and rural building stock distribution compared by income level also indicates this differences in the built environment. In low and lower middle income countries, the percentage of buildings in class I and II is indeed much higher in rural areas (36%) than in urban areas (10%). These differences are far less pronounced for higher income countries. The chosen threshold to identify rural areas in the ImageCat dataset (>50%) is larger than the average share we find in PAGER (Figure 3). This means that cells identified as rural using the ImageCat data information about the built environment with the chosen threshold are quite likely to indeed be rural.

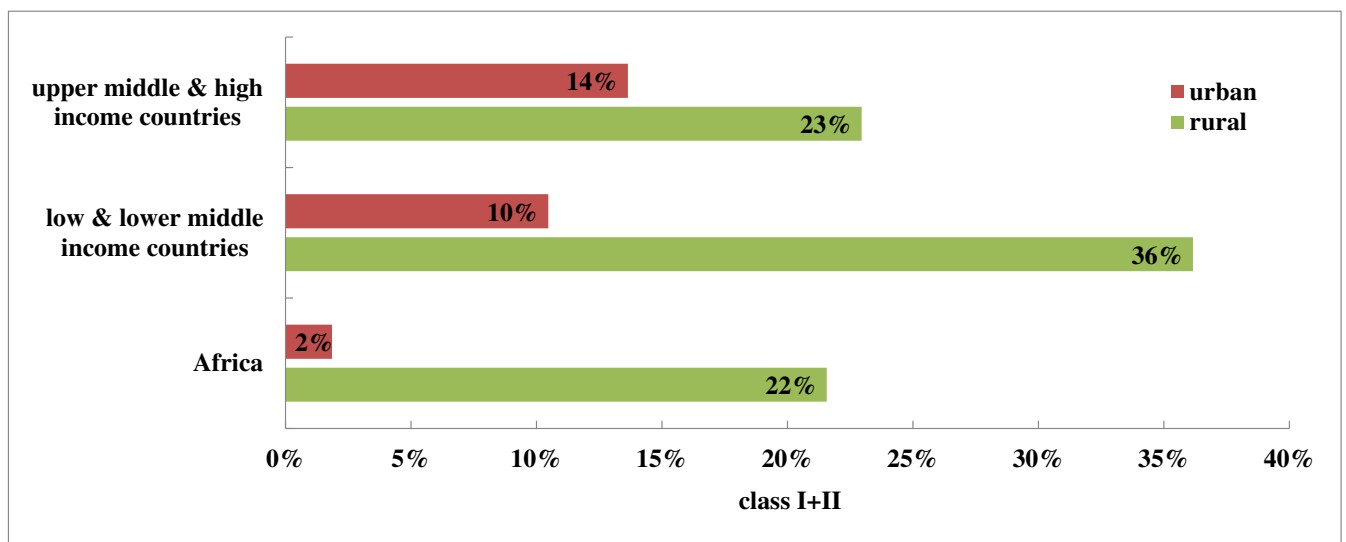

**Figure 3 Average percentage of urban and rural buildings belonging to vulnerability classes I and II for different income groups and Africa according to PAGER for countries with different urban-rural inventory.**

In remote sensing or land-use studies, accuracy assessments determine a process' accomplishment of classifying an image
(e.g. satellite data, aerial photos). Such an assessment requires reference values that represent the ground truth of the region of interest. Preferably these values are from ground collected data or hand-labelled high-resolution imagery validated by multiple interpreters (e.g. Goldblatt et al., 2018; Miyazaki et al., 2011). With these options out of the scope of this study, we examine the similarity between existing land-use products and classified areas in our approach. Compared to a strict accuracy assessment this holds the limitation of comparing already classified products. However, by benchmarking the
classified ImageCat data against established and recently published products, we provide an assessment of how well areas are identified in comparison. To this end, we reviewed the quality of the urban-rural ImageCat map by visual comparison with satellite imagery and by overlap with other classification products, visually and by quantifying the agreement between classified areas of the ImageCat data and other products (section 3.1). Two comparisons are made, one for urban and rural areas, and one only for urban areas. Similar to an accuracy assessment, we express the performance of this overlap by
calculating common comparison metrics from a confusion matrix such as overall accuracy, kappa coefficient, and producer's and user's accuracy for the sampling cells as described in Supplementary figure 1. Overall accuracy and kappa coefficient are metrics indicating the general agreement between the reference and comparison dataset. The latter two refer to the accuracy of individual classes of which the producer's accuracy describes the probability that, for example, an urban pixel is correctly classified, and the user's accuracy that a pixel classified as urban is actually urban.

For Ethiopia, the comparison maps are from several global land-use datasets as there are no other maps on national scale available for the country. For the reference map, the ImageCat data are assigned the reference categories 'urban', 'rural', and 'other land use' for cells outside of settlements. From the comparison maps, GHS-SMOD is the only other product that also considers rural settlements, allowing for a comparison of both urban and rural classifications. GHS-SMOD is a relatively

new product based on the high-resolution European Joint Research Centre's Global Human Settlement layer (Pesaresi and Freire, 2016). For GHS-SMOD, built-up areas are combined with population grids to differentiate between three settlement classes: urban centres, urban clusters, and rural (Pesaresi and Freire, 2016). In order to compare to the ImageCat reference, the GHS-SMOD's urban centre and cluster cells were reassigned into a single urban class and rural cells were kept as is.

More products are available that provide a classification limited to urban areas, but largely overlook rural areas, such as: GRUMP (CIESIN, 2011), MOD500 (Schneider et al., 2009), the Global Urban Footprint (GUF) (Esch et al., 2017), and HBASE (Global Human Built-up And Settlement Extent) (Wang et al., 2017). GRUMP and MOD500 are widely used land cover/use datasets, with GRUMP being a 30" x 30" grid of urban extent and MOD500 based on MODIS satellite data with a 500m x 500m resolution. GUF represents built-up area based on satellite imagery with a 12m x 12m spatial resolution.

HBASE is a 30m x 30m Landsat derived dataset of the extent of built-up area and settlements. All these products are used in the second comparison, in which only the 'urban' classified ImageCat settlements remain in the reference map and all cells outside of these settlements are reassigned to 'other land use'. From GHS-SMOD, the urban centre and cluster cells are again combined, but rural GHS-SMOD areas are excluded in this assessment.

Both the urban-rural and the sole urban classification comparisons between the ImageCat data and the other products follow

a class defined stratified random sampling scheme, meaning that per class 10,000 sample points were randomly placed over the cells in each reference class. As the original maps do not all share a common geospatial model, they were reprojected to a 15" x 15" raster, using the WGS-84 datum. The results of the assessments are discussed in section 3.1.

## 2.3. Maximum damage values

In step 3 (Figure 1), we determine the maximum damage of buildings in each vulnerability class. For a coherent set of input

values, we use depreciated country-specific structural maximum damage estimates per square meter as provided by the JRC report of Huizinga et al. (2017), in which residential construction costs are estimated per country using a non-linear relationship between construction costs and GDP per capita. This maximum damage value needs to be further differentiated between the four different vulnerability classes used in our study, and then multiplied by an estimate of the building footprint area per cell. This is achieved by applying the following formula for each cell:

$$D_i = \sum_1^k S \cdot N_{k,i} \cdot A_{k,i} \cdot F_k$$

Where

$D_i$ is total structural maximum damage in a given cell ($i$), $S$ is structural maximum damage per square metre in Ethiopia, $N$ is the number of buildings belonging to vulnerability class $k$ and cell $i$, $A$ is the object area, meaning the building footprint for each vulnerability class $k$ and cell $i$, and $F$ is the maximum damage adjustment factor for vulnerability class $k$.

The factors $A$ and $F$ are derived as follows:

*Building footprint area (A)*

As data on the footprint of different building types are not directly available, we estimated these based on floor area and number of floors derived from the ImageCat data. ImageCat provides estimates of floor areas for each construction type, based on sampling of building footprints, OSM data, interviews with local contractors and experts and literature review (Huyck and Eguchi, 2017). The country data descriptions also provide information on the typical number of floors, based on

sampling. For each construction type, we divided the average floor area from the ImageCat data with the number of floors, and calculated the footprint area per class ($A$) as the average from the construction types belonging to each class.

Our assumptions on the number of floors are derived from information in the ImageCat country data descriptions. Since buildings of construction types belonging to vulnerability class I or II rarely exceed one floor, we assumed them to have one floor in both urban and rural areas. The construction of class III and IV buildings with more than one floor requires a higher

skill level, mainly found in professional construction workers available in urban areas. Considering these characteristics, most class III buildings can be assumed to have one floor in rural areas. However, as most buildings in urban areas have more than one floor, we assumed class III buildings in urban areas to have two floors. Class IV buildings are assumed to be multiple floors in all areas. The buildings of class III and IV with multiple floors have a much greater footprint than the one assigned to the other classes. While buildings with smaller footprints are primarily single family residential structures or

within informal settlements, the buildings of the last two classes are mainly found in urban environments, with many of them being long apartment blocks with very large building footprints leading to a larger average footprint. The resulting building footprints for Ethiopia can be seen in Table 3.

**Table 3 Building footprints derived for Ethiopia from the ImageCat data.**

| Vuln. class | Building footprint [$m^2$] |
|---|---|
| I | 37 |
| II | 43 |
| III 1 floor | 46 |
| III 2 floors | 256 |
| IV | 467 |

*Maximum damage adjustment factor (F)*

The maximum damage values of Huizinga et al. (2017) are depreciated country-specific structural maximum damage estimates, averaged across various building types. Therefore, we differentiated these into maximum damage values for the four different vulnerability classes used in our study. Huyck and Eguchi (2017) provides estimates of replacement costs for

different structures, based on factors such as construction material and whether the structure is owner-built or engineered using professional contractors. We used these to calculate the average replacement costs for each of the four vulnerability

classes, for example the average for vulnerability class I in Ethiopia is about 95 $/sqm. In order to apply comparable maximum damage values based on a coherent dataset, these average costs per vulnerability class are then put in ratio to the country-specific values from Huizinga et al. (2017), resulting in adjustment factors ($F$) for each vulnerability class (see Table 4) to arrive at maximum damage estimates.

**Table 4 Construction cost based on Huizinga et al. (2017) and adjustment factors derived from the ImageCat data for Ethiopia.**

| Ethiopia construction costs | 671 $/sqm |
|---|---|
| Vulnerability class | Adjustment factor |
| I | 0.14 |
| II | 0.11 |
| III 1 floor | 0.18 |
| III 2 floors | 0.33 |
| IV | 0.48 |

A detailed example of the maximum damage value can be found in Supplementary figure 2. The overall Ethiopian building stock is according to the ImageCat data comprised of over 16.8mln buildings. With the described approach, the total value

exposed in urban areas amounts to about $250bln compared to almost $30bln in rural areas. Similarly, there is also a large gap between the living standard in rural and urban areas. The last Ethiopian census in 2007 (CSA, 2010) and the 2016 DHS report (CSA and ICF, 2016) provide some indications for rural and urban households that show huge differences in household durables and quality, for example more than half of the rural household with livestock share at night the room with the animals, or high quality floors in two thirds of urban households compared to only 4% of floors in rural households.

The contrasts shown there in housing characteristics such as sanitation, drinking water and flooring material illustrate that there are large differences in living conditions which indicate similar differences in exposed urban and rural value.

## 2.4. Damage and risk assessment

To calculate the damage, we combine the new exposure and vulnerability data described above, with existing hazard maps derived from the GLOFRIS global flood risk model (WRI, 2018). These maps show inundation extent and depth at a

20 horizontal resolution of 30'' x 30'' for different return periods for which per cell a Gumbel distribution was fitted to a time-series of annual maximum flood volume extracted from simulated daily flood volumes (Ward et al., 2013). Details of the original model setup of GLOFRIS are described in Ward et al. (2013) and Winsemius et al. (2013). The maps used in this study are those developed for the current time-period in Winsemius et al. (2015), which have been further benchmarked

against observations and high-resolution local models in Ward et al. (2017). In doing so, we estimate damage for the return periods 2, 5, 10, 25, 50, 100, 250, 500 and 1000 years. The inundation associated with each return period is assumed to occur everywhere simultaneously. Therefore the inundation maps are not presenting single events but country-wide probabilistic maps for the return periods. We expressed flood risk using the commonly used metric of expected annual damage (EAD).

This is estimated as the integral of the flood damage curve over all exceedance probabilities (e.g. Ward et al., 2013). A source of uncertainty in flood risk assessment is the level of incorporated flood protection. Here, we use the modelled protection standard for Ethiopia taken from the FLOPROS database, a global database of flood protection standards developed by Scussolini et al. (2016), namely 2 years.

## 3.  Results and discussion

The third chapter is organized as follows: Section 3.1 discusses the urban-rural exposure in the comparison between the ImageCat data and other products. In section 3.2, we present the results of the Ethiopian flood risk assessment using our approach and compare them in 3.3 to the results of a traditional model. In section 3.4, the sensitivity of our flood risk results is discussed for different model parameter.

### 3.1.  Urban-rural identification

The results of our classification of ImageCat cells for Ethiopia into urban or rural are shown in Table 5, along with summaries of data from other data sources. For rural areas, our result (7.2%) is similar to that of GHS-SMOD (6.4%), which is the only other data source among the products that has a specific value for rural areas. The area in Ethiopia categorized as urban or built-up is relatively low in all data sources and is in accordance with Ethiopia being one of the least urbanized countries in Sub Saharan Africa, with the share of urban population being according to Schmidt and Kedir (2009) only

between 11% and 16%, or according to more recent data from the World Bank (2016) at about 20%.

**Table 5 Cell areal extent of different land-use categories in Ethiopia as a percentage of the country area according to different products (original dataset projections).**

| Dataset | % of country |
| --- | --- |
| ImageCat | urban 0.6%, rural 7.2% |
| GHS-SMOD | urban centre 0.4%, urban clusters 1.1%, rural 6.4% |
| GRUMP | urban extent 0.5% |
| MOD500 | urban extent 0.1% |
| GUF | built-up area 0.1% |
| HBASE | built-up area and settlements 0.1% |

*Visual comparison*

Our urban-rural classification is shown spatially in the example of Figure 4, in which we compare different land-use products for an area near the City of Awasa. The urban and rural areas identified in GHS-SMOD and our classified ImageCat data show a more detailed and differentiated representation of the settlements than the coarse resolution GRUMP

and MOD500 products. All products overlap in the location of main urban areas, although their extent varies. Locations of built-up areas with medium extent, for example in GUF, are hardly or not detected in HBASE, MOD500, and GRUMP, but are also seen with GHS-SMOD and our ImageCat classification.

Using our classification method, some smaller settlements are labelled urban with the ImageCat data, because their building stocks have high shares of class III and IV buildings, whilst GHS-SMOD classifies them as urban clusters or rural. Examples

are the areas around Shashemene (see circled examples in Figure 4a). By visual inspection of Google Earth, these seem to be areas of urban-rural transition. They have a more densely built environment than rural areas and a higher number of class III and IV buildings, which leads to the urban classification in our method. Areas where cells from the ImageCat data get classified as rural are also rural in GHS-SMOD or to some extent urban clusters due to a higher population density in the surrounding cells. However, the overlap of these settlements is more about the general area and less regarding a cell by cell

comparison. In addition, visual inspection showed that the small, more widespread settlements such as east of Awasa and Shashemene are correctly detected in the ImageCat data (rural areas in Figure 4a) but are not displayed in GHS-SMOD (Figure 4b). As a consequence of these issues, it is expected that the classified ImageCat data and GHS-SMOD overlap is lower for rural than urban settlements.

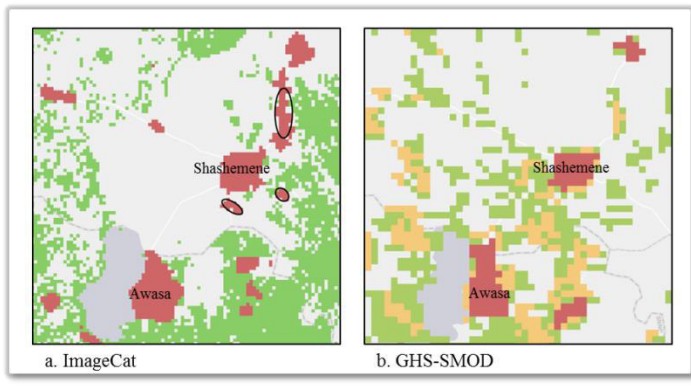

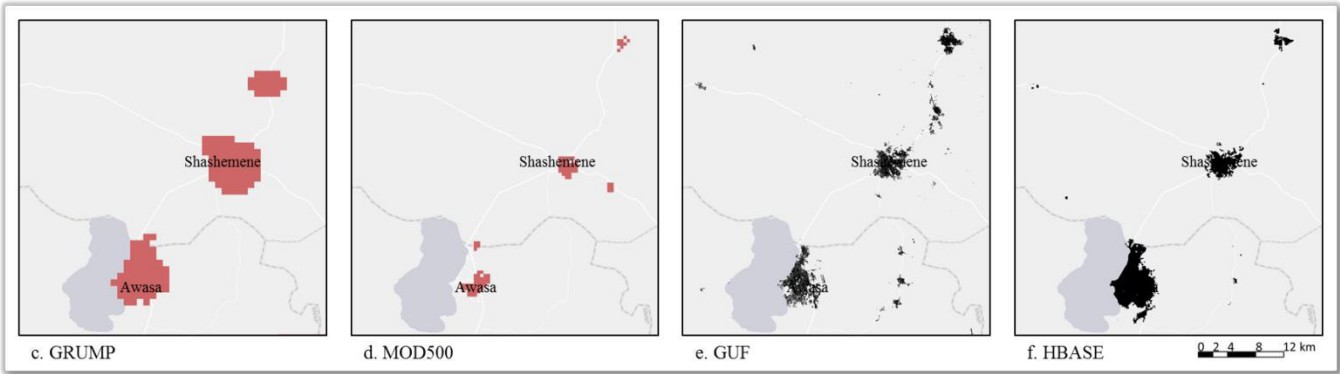

**Figure 4 Illustration of urban-rural land use in the greater Awasa area in Ethiopia: (a) Urban (red) and rural (green) classified ImageCat data, (b) GHS-SMOD urban centre (red), urban cluster (yellow), rural (green), (c) GRUMP urban extent (red), (d) MOD500 urban extent (red), (e) GUF built-up area (black), (f) HBASE built-up area and settlements (black); original dataset projections.**

*Map agreement analyses*

Map agreement has been assessed for urban-other classes, and urban-rural-other classes using confusion matrices (see supplementary table 2 and supplementary table 3). When comparing the urban areas (supplementary table 4), we see that urban and built-up area cells in the GRUMP, MOD500, GUF and HBASE almost always correspond with urban cells in the ImageCat map (urban user's accuracy ~99-100%). This confirms the observations from the visual comparison (Figure 4) where we see that the general location of the main urban areas are similar between the datasets. However, with the ImageCat data more medium-sized urban areas are detected which are often not in the other datasets, resulting in the low producer's accuracy (~6-26%), again confirming the visual comparison of the Awasa region.

When including rural settlements in the assessment, only GHS-SMOD and the ImageCat classification can be compared (Table 6), as they are the only datasets which distinguish rural areas. This comparison is complicated by the fact that GHS-SMOD has three categories (urban centres, urban clusters and rural). Visual comparison with satellite imagery reveals that the middle class of urban clusters are sometimes an extension of urban centres, but can also refer to higher density settlements areas in rural areas. Nevertheless, for the map agreement analysis of urban-rural-other classes we grouped these

urban clusters with the urban centres to form the urban class. We find that urban cells in the GHS-SMOD have a high probability to also be urban areas in the ImageCat map (urban user's accuracy of 86.3%). However, urban cells from the ImageCat data have a much lower probability to be urban in GHS-SMOD (urban producer's accuracy of 48.7%). This implies that there are various urban settlements in the ImageCat map, which are not present in the urban group (centres and
clusters) of the GHS-SMOD.

The agreement of rural cells is less good as compared to the urban cells, with considerably lower user's and producer's accuracies (31.3% and 11.0% respectively). Classifications of the built-up land from remote sensing based products inherently have lower accuracy levels in less developed regions and rural settings. Even high resolution products still omit large shares of built-up areas and have to improve their performance in arid regions of Africa and areas where settlements
are more scattered (Klotz et al., 2016; Leyk et al., 2018). We can also observe this in the visual comparison (Figure 4) where the high resolution GUF and HBASE datasets omit many of the scattered settlements that are found in the ImageCat data or GHS-SMOD. Because of these difficulties in detecting such scattered settlements, the agreement between rural areas from the ImageCat classification and in GHS-SMOD is adversely affected as one dataset might indicate rural areas that are not identified in the other.

Comparability of classified maps remains an issue. For example, it has been illustrated in the literature that the total urban land in global maps varies by an order of magnitude between early global earth observation products and GRUMP. Likewise, there is about a factor 5 difference between MOD500 and GRUMP (Potere et al., 2009), and the global built-up area in the high resolution GUF product is 35% less than in GHS built-up (Esch et al., 2017). ImageCat data is more specific to the African context as the other maps are based on global classification algorithms.

The construction types based ImageCat classification is a distinctly different approach as compared to most classifications, which use population and/or built-up densities. This can also cause some mismatches, for instance in informal settlements in or around cities which are classified as urban when looking at densities, but would be classified as rural when looking at construction types. Our analysis showed, however, that the classification from ImageCat data is overall reasonably similar to existing datasets, and it includes unlike other land-use products rural settlements, and as such a good alternative for flood
risk assessments as it provides the option for more detailed building-material-based vulnerability curves in the analysis.

**Table 6 Results of map agreement for Ethiopia using the ImageCat data classified to urban, rural, and other land use as the reference map.**

| Urban-Rural Map | Urban | | Rural | | Other land use | | Overall Accuracy (%) | Kappa |
|---|---|---|---|---|---|---|---|---|
| | Producer's Accuracy (%) | User's Accuracy (%) | Producer's Accuracy (%) | User's Accuracy (%) | Producer's Accuracy (%) | User's Accuracy (%) | | |
| GHS-SMOD | 48.7 | 86.3 | 11.0 | 31.3 | 94.8 | 45.5 | 51.5 | 0.27 |

### 3.2. Flood risk assessment

Modelled flood damage for the different return periods and risk for urban and rural areas are shown in Figure 5. To calculate the overall risk in the country, these simulation are based on probabilistic maps for which inundation associated with 2, 5, 10, 25, 50, 100, 250, 500 or 1000-year return period respectively occurs simultaneously in all flood affected cells. For 2-year
return periods the damage is always zero as it is assumed that these floods would not cause overbank flooding. As can be expected, the damage in urban areas is higher, as it is a more densely concentrated built-up environment and the value of the buildings is higher. On the other hand, the majority of exposed buildings are in rural areas. To illustrate, about 88,000 buildings in urban areas of Ethiopia are exposed to a flood of a 100-year return period, compared to more than four times as many rural buildings. Furthermore, we can see that large damage already occurs for higher probability flooding, for example
for the 25-year return period flooding the country-wide rural damage already amounts to over \$200mln and over \$700mln for damage in urban areas.

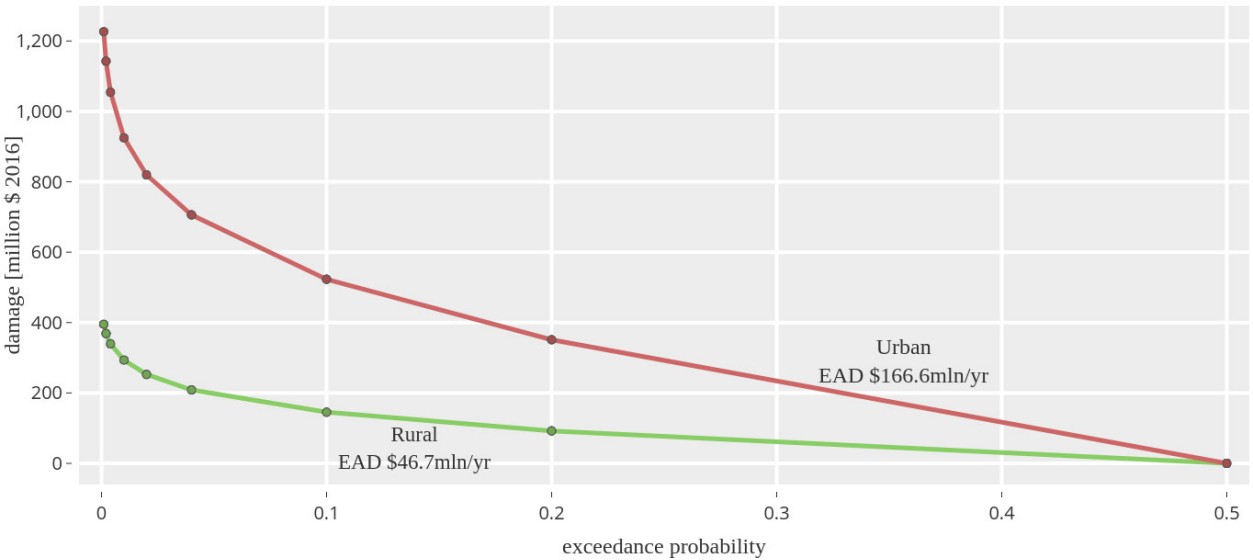

**Figure 5 Risk curve for simulated flood damage to building structures in urban and rural areas of Ethiopia for return periods from 2 to 1,000 years.**

Table 7 shows the EAD for the different vulnerability classes in urban and rural areas. These results show that most of the damage in rural areas results from damage to buildings of class I, which are buildings with the highest vulnerability. In urban areas, the largest share of the damage results from damage to buildings of class IV; these are the buildings with the highest exposed values. In addition, this class also makes up a large share of the exposed urban buildings, about 57,000 for a flood of a 100-year return period which is more than twice as many buildings of class III. In total more than 464,000 buildings are
simulated to be affected for flooding with this return period, but most are in rural areas with the majority belonging to class I (58.3%) (class II 14.6%, class III 8.1%).

**Table 7 Expected annual damage (in Million $ 2016) to building structures by vulnerability class in urban and rural areas of Ethiopia.**

|       | I    | II  | III  | IV    | all   |
|-------|------|-----|------|-------|-------|
| Rural | 31.1 | 8.3 | 7.3  | 0     | 46.7  |
| Urban | 0.3  | 0.2 | 29.8 | 136.2 | 166.6 |
| Total | 31.4 | 8.5 | 37.1 | 136.2 | 213.2 |

The overall flood risk in Ethiopia (i.e. expected annual damage, EAD), is about $213.2mln/yr; 78% of the total EAD is in urban areas. Whilst the rural EAD is below the EAD in urban areas, it is still high in absolute terms ($46.7mln/yr). This demonstrates that neglecting damage to rural buildings in large-scale assessments may lead to a severe underestimation of total damage values. Furthermore, the flood damage in urban and rural areas have to be considered in the context of the coping capacity of the population in the respective areas. The flood vulnerability of people below the poverty line is higher, as a larger proportion of their wealth could be affected during a flood event (Winsemius et al., 2018). While this is also true for the urban poor, the livelihoods of rural people are more susceptible where services and infrastructure are limited (Komi et al., 2016).

### 3.3. Comparison with Aqueduct

Compared to a traditional land-use-based model, the total simulated damage in our approach is somewhat higher, but similar in magnitude. For example, the new version of the GLOFRIS model used for the Aqueduct Global Floods tool (WRI, 2018) applies the same inundation data as used in this study, but follows the common approach of using land-use-based exposure and vulnerability data, resulting in EAD for Ethiopia of $182mln/yr. The results from our approach contain much more detail on the exposed elements and their vulnerability and allow us to examine damage in urban and rural areas. Damage in urban and rural areas cannot be distinguished in GLOFRIS as it uses HYDE data (Klein Goldewijk et al., 2011) to represent exposure, which represents the urban built-up fraction per grid cell. Moreover, Figure 6 compares the land use exposure map using classified ImageCat data and HYDE for the example of Addis Ababa. As for the rest of the country, it demonstrates that datasets like the ImageCat exposure data can provide more spatial detail than the commonly used exposure maps such as HYDE used in land-use-based flood risk models. Settlement extent and outlines are more distinctive, resulting in an overall better representation of affected settlement areas in the risk assessment of our approach.

Further risk comparison as well as flood protection influence can be found in supplementary section 2.

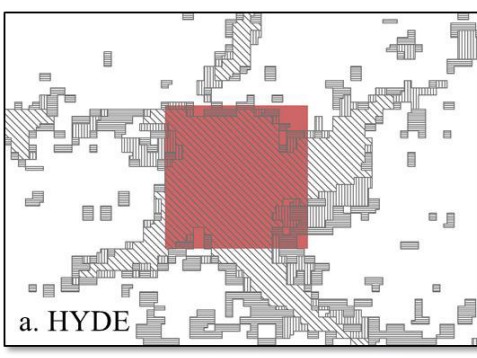

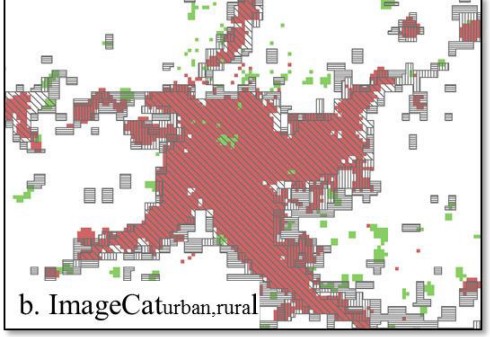

**Figure 6 Addis Ababa mapped by a. HYDE as used in GLOFRIS with above 0% urban built-up (red); b. classified ImageCat data urban (red), rural (green); GHS-SMOD rural (horizontal), urban cluster (vertical), urban centre (diagonal) as background boundary reference.**

### 3.4. Sensitivity analysis

Given the uncertainty in the input datasets and methods used in our approach, we perform a one-at-a-time sensitivity analysis to assess how the simulated EAD is affected by our assumptions on the: (a) structural maximum damage values; (b) threshold used in the urban/rural classification; (c) object area; and (d) stage-damage curves.

To assess the sensitivity of the results to the assumed values for maximum damage, we used the 90% confidence interval of estimated construction costs for residential buildings reported by Huizinga et al. (2017). These state that construction costs

can be between 28% lower and 53% higher than the estimates used in this paper. For sensitivity to the threshold used in the urban/rural classification, we used thresholds of 20% and 80% for classifying urban areas, instead of the 50% used in the earlier analysis. Object areas can be very diverse between and within countries and depend on the characteristics of the housing market. For example, the Centre for Affordable Housing Finance in Africa yearbooks include some indication on the average house size and price per country. However, the used sample sizes for example are very small and the average

value covers only the minimum size that formal developers in urban areas are prepared to build, therefore neglecting self-built houses. Furthermore, no differentiation between building types or constructions is given (CAHF, 2017). For the

sensitivity analysis, instead of calculating the footprint areas from average floor areas across the construction types per vulnerability class, we used the most frequent floor area size per type in the ImageCat data. The building footprint sizes most affected by this are those for classes II and III (see supplementary table 5), as the size decreased by 5 to 11m$^2$. The state-damage curves in this study show a wide range of vulnerability (see Figure 2). Nonetheless, this as well as a comparable shape can also be found in the for different continents identified residential curves by Huizinga et al. (2017) as for example their damage ratios at 1m range between 38% to 71%. While our vulnerability functions show high degrees of damage particularly for class I and II (mud/adobe and wooden buildings), other functions that consider building structure such as in the CAPRA project (CAPRA, 2012; Wright, 2016) display similar behaviour for these types of buildings. The sensitivity regarding the vulnerability curves is analysed by applying like most traditional flood risk models only one vulnerability curve, thus neglecting the differentiation our model makes toward material-based vulnerability. To this end, we selected the residential stage-damage curve used in GLOFRIS, for which the degree of damage progresses slightly below the class III one floor curve.

**Table 8 Expected annual damage (in Million $ 2016) compared for the normal model setup and the modified parameters used in the sensitivity analysis.**

| | Normal Run | Sensitivity Analysis | | | | | |
| | | Max. Damage | | Urban-Rural | | Object Area | Vuln. Curve |
| | | lower | upper | 20% | 80% | | |
|---|---|---|---|---|---|---|---|
| Rural | 46.7 | 33.6 | 71.4 | 46.7 | 46.7 | 41.5 | 37.4 |
| Urban | 166.6 | 119.9 | 254.8 | 166.6 | 166.6 | 165.8 | 264.1 |
| Total | 213.2 | 153.5 | 326.2 | 213.2 | 213.2 | 207.3 | 301.5 |

Results of the sensitivity analysis are summarised in Table 8. Clearly, the flood risk estimate is very sensitive to the applied maximum damage values, as the EAD scales linearly with maximum damage changes. The results also show the EAD to be sensitive to the applied vulnerability curve. Using the single curve from GLOFRIS leads to a higher total estimate of risk by 41%. Therefore, the estimation of maximum damage values and improved representation of vulnerability are important considerations for large-scale flood risk modelling. Our approach improves the incorporation of vulnerability in the risk assessment by differentiating a built environment into classes that characterise the vulnerability of a building stock even on large scales. The EAD is very insensitive to the threshold used in the urban/rural classification. Even with the wide range of thresholds used in the sensitivity analysis, influence on the urban-rural distribution is minimal, confirming that the urban and rural built environment in Ethiopia is very distinct in terms of what materials and construction types are applied. Nonetheless, as previously discussed in section 3.1, exposure of an area can vary depending on the applied dataset. Using ImageCat data, over half of the construction types in Ethiopia belong to class I, and about 14% towards each of the other

classes (see Table 9). However, according to data from the last census in Ethiopia from 2007, 73.9% of all housing units in Ethiopia have been assigned the 'wood and mud' wall material, with 80% of the urban units and 72.5% of rural units, whereas a large share of rural units were built with wood (and thatch) walls (15.5%). Compared to the ImageCat data, the Ethiopian building stock appears to be less diverse and shows a different distribution of urban and rural constructions, which is also affected by the applied definition of urban in the census. Since the 2007 census, Ethiopia has experienced considerable economic growth that appears to coincide with growth in the Ethiopian construction industry (World Bank, 2019). Furthermore, census data are aggregated to administrative levels and thus cannot be applied in the approach developed in this paper, for which an object-based dataset is required that is comparable between countries, such as the ImageCat data. With different methodologies in exposure datasets, future research should explore how flood risk assessments that are based on building-material-based vulnerability are affected by the applied building stock dataset and their different scales. In our sensitivity analysis, the assumptions made on the object areas have little influence on the EAD, with overall slightly lower EAD when using alternative footprint sizes. Even though the effect of the object areas is small, it must be noted that these are estimated sizes and in reality building layouts are very diverse.

**Table 9 Ethiopian building stock according to ImageCat data**

| Type | Description | % total building stock | Class | % urban building stock | % rural building stock |
|------|-------------|------------------------|-------|------------------------|------------------------|
| ADB | URM adobe building | 4.1 | | | |
| ERTH | Earthen building | 3.9 | I | 3.4 | 72.0 |
| INF | Informal building | 9.4 | | | |
| WWD | Wattle & daub building | 39.7 | | | |
| WLI | Light wood building | 1.0 | II | 2.0 | 18.0 |
| WS | Solid wood building | 13.5 | | | |
| BRK | URM brick building | 6.1 | III | 29.9 | 10.0 |
| STN | URM stone building | 8.2 | | | |
| RC | Reinforced concrete frame with URM infill building | 13.9 | IV | 64.8 | 0.03 |

## 4. Conclusions and recommendations

In this paper, we investigated how characteristics of the built environment can be used to assess flood impacts on large scales. To this end, we developed flood vulnerability classes and stage-damage curves that are based on construction types and building materials. In contrast to other large-scale flood risk models that employ aggregated land-use categories and

vulnerability curves, our approach takes advantage of detailed information of the exposed elements to differentiate their vulnerability.

Showing that the predominant types of buildings are different in urban and rural areas, particularly in developing countries, the settlements' land use can be identified by the characteristics of their building stock. By distinguishing the urban and rural

built environment using our vulnerability classes, we opened up the possibility to analyse flood impacts outside of the typical focus on urban areas of large-scale flood assessments. We used it to show how flood damage to buildings differ and assessed flood risk in urban and rural areas of Ethiopia. Although EAD in urban areas exceeds EAD in rural areas, the rural flood risk of $46.7mln/yr (over 20% of total risk) is nevertheless significant. Moreover, far more buildings are affected in rural as opposed to urban areas. As low water depths can already cause major damage to the types of buildings that predominantly

exist in rural settings in Africa, differentiation between flood damage in urban and rural settings could also be invaluable to studies related to poverty and flooding.

We examined the effects of parameter uncertainty and found that the model is insensitive to the applied threshold identifying urban and rural areas from the object-based information about the characteristics of building stock in the study area using our material-based vulnerability classes. Consistent with other studies (e.g. de Moel and Aerts, 2010; Merz et al., 2010), the

sensitivity analysis showed that the replacement value of the exposed buildings deserves considerable attention as we see large differences in the model output. The results further showed that aggregated vulnerability as used in large-scale land-use-based models affects the results to a great extent. In our model, vulnerability is addressed in greater detail as it reflects the behaviour of different types of buildings during floods according to their structural characteristics. Therefore, it provides a more direct relation between physical damaging processes and flood impact on different structural types.

This approach is of particular importance for studies where there is a large variation in construction types, such as large-scale studies or studies in developing countries for which the urban and rural building stock is much more differentiated. Large informal settlement areas in cities are not specifically addressed in the current setup and would be classified as rural. To acknowledge this, the urban-rural classification could be extended to highlight such areas and ones where none of the typically urban or rural building types clearly prevail. Lastly, it has to be noted that maintenance can influence the quality of

the construction over the years, thus the structural vulnerability would further increase with building age. Future research would benefit including these indicators or similar ones such as building laws and practices, given that sufficient data becomes available, to highlight differences between regions. Furthermore, if the data allows in the future, vulnerabilities within the classes could be further refined such as between clay, stone and concrete brick/block construction or regarding non-structural elements like electrical components and partition walls.

Besides improving the accuracy in estimating direct flood damage, the use of building-material-based vulnerability curves also paves the road to the enhancement of multi-risk assessments as the method enables the comparison of vulnerability across different natural hazard types that also use building-material-based vulnerability.

## Acknowledgements

This work was supported by the Netherlands Organisation for Scientific Research (NWO) in the form of VICI grant 453.140.006 for JCJHA and VIDI grant 016.161.324 for PJW. The ImageCat exposure data is based on work supported by the National Aeronautics and Space Administration under Grant NNX14AQ13G, issued through the Research Opportunities in Space and Earth Sciences (ROSES) Applied Sciences Program. The views and conclusions contained in this presentation are solely those of the authors.

## Competing interests

The authors declare that they have no conflict of interest.

## Author contribution

JE, HdM, and PJW conceived the study. JE, HdM, and MCdR developed the vulnerability classification and conducted the literature review. The methodology was designed by JCJHA, JE, HdM, and PJW, with exposure data provided by ImageCat and CKH contributing to the enrichment of the analysis and discussion of results. JE analysed the data and prepared the draft, with all co-authors commenting on the manuscript.

## Data availability

This work relied on data which are available from the providers cited in section 2 and 3.

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
