# Peer review of "Enhancement of large-scale flood risk assessments using buildingmaterial-based vulnerability curves for an object-based approach in urban and rural areas"

_Natural Hazards and Earth System Sciences, 2019_

## Referee Comment (RC1) · Anonymous Referee #1 · 18 Apr 2019

Referee report on NHESS-2019-32, "Enhancement of large-scale flood damage assessments using building-material-based vulnerability curves for an object-based approach"

The authors present an approach for a nation-wide (large-scale) flood risk assessment based on available data on hazard frequency and magnitude, downscaled vulnerability information to allow for an object-based assessment, and corresponding information on elements at risk. As such, their approach is timely, and allows for computation of risk. The work will be of considerable importance to the readers of NHESS. Therefore, I recommend acceptance of the Discussion manuscript to NHESS, given some clarifications outlined below.

First of all, I recommend to adjuste the title to better reflect the content of the work, examples include "Enhancement of nation-wide flood risk assessment using building-specific vulnerability curves for an object-based approach" because (a) large-scale could be misleading within the geo community (1:1 = large scale, 1:1,000,000 = small-scale) and (b) the overall framework presented in the introduction of this work is related to risk, not damage (or loss) assessment. Moreover, even "Africa" could be included in the title.

Second, I would remove the sentence on "multi-risk" from the Abstract since the paper focuses on flood risk. With respect to the use of this term, however, the authors might wish to include some sentences on flood characteristics in Ethiopia, as in many countries with semi-arid climate "flood" may also contain fast-onset processes such as flash floods which are in their characteristics and assessment different from "traditional river flooding", and I was not sure whether or not these types of hazards are also included in the dataset used for hazard assessment in the study.

Third, in the introduction page 2, line 1 f. the authors include their risk concept following the UNISDR definition. In the third paragraph, in contrast, the authors are reporting on a vulnerability curve which "is generally developed for each of the aggregated land-use categories used to represent exposure (Ward et al., 2013)" – this is contradictory to the above-mentioned definition, and is also not followed throughout the paper. The authors also mentioned this challenge on page 2, lines 21-25. Maybe here also some sentences on the conceptualization of "vulnerability" may be useful, as the paper deals with what is named "physical vulnerability" in many works, in contrast to e.g. "social" and "institutional" vulnerability.

Fourth, the paragraph on literature related to "flood" vulnerability of buildings would surely also gain from a more thorough distinction between different flood types (such

as the above-mentioned flash floods), examples may be found in recent NHESS publications (and not only these from the [Dutch] flood communities). In particular with respect to the challenge of structural vulnerability, other scholarly articles may be found in the Journal of Hydrology, Geomorphology, Engineering Geology, or Water Resources Research, all of them focusing on the consideration of the mentioned building type, quality, height, and material during vulnerability assessment. This could be used to expand the statement made on page 3, lines 14 ff.: "Compared to risk assessments in the earthquake domain where they are essential components (de Ruiter et al., 2017), or in 15 local-scale studies focusing on physical vulnerability to debris flows (Papathoma-Köhle et al., 2017), construction types and building materials have only played a minor role as indicators for flood vulnerability. Large-scale flood risk assessments could be improved by using object-based characteristics to represent exposure and vulnerability [. . .]"; examples may include Milanesi et al. (2018), Sturm et al. (2018a, b), Zhang et al. (2016; 2018), Kang and Kim (2016), Godfrey et al. (2015), or Thouret et al. (2014).

Fifth, with respect to Table 1 and the related text body I was wondering if and how results from different country setting such as e.g., Germany, Japan, China and Malawi could be better combined – in many countries building laws, design criteria, construction types and technical knowledge as well as economic feasibilities are different, leading to significantly different resistance to flood hazards (independent of which flood type). So the authors could better explain how they concluded or deduced their classification scheme of four building classes (which in practice is good), maybe by only focusing on the situation in Africa. Elsewhere, the authors cite the recent review of Papathoma-Köhle et al. (2017), where a main conclusion was that "environmental, as well as the socio-economic context of areas subject to (. . .)[floods] varies around the world. A one size fits all method is difficult to be developed, on the other hand, investing time and effort in the development of tools that are tailor made only for a specific area is also counterproductive. Methodologies may be transferred to other areas, however, this is not always the case for the final tools (e.g. a vulnerability curves). To which extent a vulnerability curve or a specific weighting of indicators is transferable to another

area has to be further investigated."

Sixth, I would like to recommend a discussion on the vulnerability curves shown in Figure 2 and used within the present study. Their shape is considerably different from the shape of either traditional flood loss models (e.g., Kreibich et al., 2010) but also from models used in flash-flood vulnerability assessments such as those presented by Karagiorgos et al. (2016) or even from hydrological hazards in mountain areas (such as e.g., Totschnig and Fuchs 2013). Moreover, the curves provided in Figure 2 clearly show that the main damage already occurs with relatively small flood intensities up to 1.0 m, which could be worth to discuss further – as such, the use of a 1:1,000 year flood hazard map seems a bit over-ambiguous if we assess recent flood events over the African continent (already 1:50 to 1:100 year events are responsible for high loss rates and those should be better included in any hazard and risk mitigation strategy); moreover, the mentioned informal settlements are very often located in flood plains which are even affected by larger frequencies.

Seventh, even if the authors discussed sources of uncertainty for every risk factor used during the set of calculations, it would be great to have a summarizing chapter on the overall sources of uncertainty and spread (maybe including an overview Table) – such as e.g. discussed with respect to Figure 4, there are many sources of uncertainty to be considered when applying "large-scale" nation-wide risk assessment (such as e.g. discussed on page 16, lines 20 ff.: "…varies by an order of magnitude", or further down in lines 26 ff.: "…mismatches, for instance in informal settlements…". This issues is also shown on page 20, lines 14 ff: "Using the single curve from GLOFRIS leads to a higher total estimate of risk by 41%. Therefore, the correct [btw: what exactly do you mean with "correct"?] estimation of maximum damage values and improved representation of vulnerability are important considerations for large-scale flood risk modelling." Other examples include page 21, lines 18 ff: "Consistent with other studies (…), the sensitivity analysis showed that the value of the exposed buildings deserves considerable attention as we see large differences in the model output. The results

further showed that aggregated vulnerability as used in large-scale land-use-based models affects the results to a great extent."

Some small items:

- Please carefully check the use of "damages" versus "damage" since in the insurance industry, "damages" is used slightly different.

- Page 2. Line 30: "therefore" instead of "therefor"

I think that considering these items will result in a more concise presentation of methods and results, and will considerably improve the valuable study presented in NHESSD. Therefore I kindly encourage the authors to proceed with their works, and undertake these revisions.

Please note that the references cited in this review are for clarification and illustration purpose only, the decision which to include in a revised version shall definitely be with the authors of this NHESS manuscript.

References mentioned

Godfrey, A., Ciurean, R. L., Van Westen, C. J., Kingma, N. C., and Glade, T.: Assessing vulnerability of buildings to hydro-meteorological hazards using an expert based approach – an application in Nehoiu Valley, Romania, International Journal of Disaster Risk Reduction, 13, 229-241, https://doi.org/10.1016/j.ijdrr.2015.06.001, 2015.

Kang, H., and Kim, Y.: The physical vulnerability of different types of building structure to debris flow events, Natural Hazards, 80, 1475-1493, https://doi.org/10.1007/s11069-015-2032-z, 2016.

Karagiorgos, K., Thaler, T., Heiser, M., Hübl, J., and Fuchs, S.: Integrated flash flood vulnerability assessment: insights from East Attica, Greece, Journal of Hydrology, 541, 553-562, https://doi.org/10.1016/j.jhydrol.2016.02.052, 2016.

Kreibich, H., Seifert, I., Merz, B., and Thieken, A. H.: Development of FLEMOcs –

a new model for the estimation of flood losses in the commercial sector, Hydrological Sciences Journal, 55, 1302-1314, https://doi.org/10.1080/02626667.2010.529815, 2010.

Milanesi, L., Pilotti, M., Belleri, A., Marini, A., and Fuchs, S.: Vulnerability to flash floods: A simplified structural model for masonry buildings, Water Resources Research, 54, 7177-7197, https://doi.org/10.1029/2018WR022577, 2018.

Papathoma-Köhle, M., Gems, B., Sturm, M., and Fuchs, S.: Matrices, curves and indicators: a review of approaches to assess physical vulnerability to debris flows, Earth-Science Reviews, 171, 272-288, https://doi.org/10.1016/j.earscirev.2017.06.007, 2017.

Sturm, M., Gems, B., Keller, F., Mazzorana, B., Fuchs, S., Papathoma-Köhle, M., and Aufleger, M.: Experimental analyses of impact forces on buildings exposed to fluvial hazards, Journal of Hydrology, 565, 1-13, https://doi.org/10.1016/j.jhydrol.2018.07.070, 2018a.

Sturm, M., Gems, B., Keller, F., Mazzorana, B., Fuchs, S., Papathoma-Köhle, M., and Aufleger, M.: Understanding the dynamics of impacts at buildings caused by fluviatile sediment transport processes, Geomorphology, 321, 45-59, https://doi.org/10.1016/j.geomorph.2018.08.016, 2018b.

Thouret, J.-C., Ettinger, S., Guitton, M., Santoni, O., Magill, C., Martelli, K., Zuccaro, G., Revilla, V., Charca, J. A., and Arguedas, A.: Assessing physical vulnerability in large cities exposed to flash floods and debris flows: the case of Arequipa (Peru), Natural Hazards, 73, 1771-1815, https://doi.org/10.1007/s11069-014-1172-x, 2014.

Totschnig, R., and Fuchs, S.: Mountain torrents: quantifying vulnerability and assessing uncertainties, Engineering Geology, 155, 31-44, https://doi.org/10.1016/j.enggeo.2012.12.019, 2013.

Zhang, J., Guo, Z. X., Wang, D., and Qian, H.: The quantitative estimation of the

vulnerability of brick and concrete wall impacted by an experimental boulder, Natural Hazards and Earth System Sciences, 16, 299-309, https://doi.org/10.5194/nhess-16-299-2016, 2016.

Zhang, S., Zhang, L., Li, X., and Xu, Q.: Physical vulnerability models for assessing building damage by debris flows, Engineering Geology, 247, 145-158, https://doi.org/10.1016/j.enggeo.2018.10.017, 2018.

---

## Referee Comment (RC2) · Anonymous Referee #2 · 24 Apr 2019

This is a very good paper, focusing on the importance of using building-material-based information in the exposure, vulnerability components of large-scale (global) flood modelling efforts.

The paper demonstrates clearly how such work is making significant improvements in flood risk assessment. Another important part is the discussion of spatial capture of urban-rural areas. This merits to also be included in the paper's title.

My review focused more on this aspect of the paper's content. Please see my com-

ments in the attached PDF file.

I am concerned that the estimation of the replacement value of the buildings in Ethiopia shows a big urban-rural divide (buildings per capita exposure being 32 times greater in the urban areas vs the rural areas).

Since this paper is applying the proposed methodology to Ethiopia it is very important to use Ethiopia data. It is necessary to revise the entire section "Object-based exposure data" to include review of the 2007 Ethiopia census.

Once this is done it will be also apparent that the section "3.2. Flood risk assessment" also needs to be revised because the building stock distributions of classes I to IV in Ethiopia are quite different to what the authors have probably assumed. In this section a Table of classes I, II, III, III2 and IV distribution of the building footprints in urban and rural Ethiopia used in the model is not shown and this is an important omission.

This part of the work, i.e. the passing from census data to classification of the building vulnerability classes and the building footprints needs to be much more clearly explained than it is in the present version with some additional references for the ImageCat methodology.

Please also note the supplement to this comment:
https://www.nat-hazards-earth-syst-sci-discuss.net/nhess-2019-32/nhess-2019-32-RC2-supplement.pdf

**Supplement:**

[revised manuscript text omitted]

**Comment [A3]:** In the 2007 census of Ethiopia the most common wall-type is "Mud and Wood" forming 80% of houses in Urban & 72.5% in Rural. In rural areas the next most common are "Wood and Thatch / Wood only" (15.5%).

[Figure]

flood depth of 2.5 m (i.e. damage can reach full building value, unlike masonry and concrete constructions). Buildings that are based on wood construction types can account for a large proportion of overall building stock in some countries (e.g. USA, Japan and Ethiopia). The quality of these constructions and the building's value can vary considerably. For large-scale assessments outside of Africa, adjustment towards a greater flood resistance is recommended.

*Class III* are unreinforced masonry/concrete buildings with walls of burnt clay bricks or stone or concrete blocks. These buildings are more vulnerable than those in class IV (reinforced masonry/concrete or steel). This is related to the fact that unreinforced walls are less able to resist the pressure of flood water exerted on the load-bearing walls. However, damage potential is assumed to be less than class II, as masonry clay bricks, stone and concrete blocks are more durable and less likely to disintegrate or need

replacement after being flooded compared to wood. Therefore, a curve between class II and class IV was created for both one and two storey buildings of this class.

*Class IV* represents engineered reinforced masonry/concrete and steel buildings. These types of buildings are engineered and basically standard in most western countries and large cities in Africa. Overall, they constitute the most resistant class to flooding. Many studies (e.g. Buck,

2007; Li et al., 2016; Maiti, 2007) show that vulnerability curves for these types of buildings do not go up to a damage factor of 1, as some elements do not need replacement after a flood (e.g. the foundation or  the structural walls or the frames). This is similar to the values from Dutta et al. (2003) and HAZUS-MH (Scawthorn et al., 2006), who show examples of curves that go up to 0.6-0.7 damage ratio. Therefore, in this study it is chosen to let this curve go up to 0.65. Both reinforced masonry and reinforced concrete and steel are put in the same class IV.

> **Comment [A4]:** In future, if the data allow, differentiating vulnerability between clay bricks, stones and concrete blocks should be considered.

> **Comment [A5]:** These buildings tend to have more non-structural elements that can be vulnerable to flooding especially in Africa's Urban areas e.g. air conditioning units, partition walls, mechanical & electrical components, etc. that would need to be considered both in terms of their contribution to the overall building replacement value and their vulnerability. At a future stage.

[revised manuscript text omitted]

**Comment [A14]:** In PAGER for Africa there is the problem that only 19 of the 56 countries have original data (and of these 6 are from 1993), the rest are based on "neighbor country". Also these data are primarily distributions of the housing units, not the Residential + Non-Residential buildings, and in urban areas the building distributions would be quite different due to many houses being in apartment buildings.  Also differentiation for Urban-Rural and Residential-Non-Residential exists only for 2 countries (Algeria & Morocco). The value of 2% in Urban is for Algeria. In rural Algeria this value is 15%.
In the 2007 Ethiopia Housing census the ratio of class I & II in Urban is 89% (81% in Addis Ababa) which would challenge the rural hypothesis (>50%)..
Since this paper is examining Ethiopia it would be better to use the data from the 2007 Ethiopia Census (also available in PAGER v2) that gives distributions of the housing units in urban and rural areas or use the distributions of Ethiopia in PAGER (though they do not differentiate urban-rural).

**Comment [A15]:** Add in the Supplementary References: Congalon, 1991

new product

[revised manuscript text omitted]

**Comment [A19]:** These values when summed would suggest that the replacement value of Ethiopia's building stock is assessed at 384% of 2016 GDP. The per capita buildings exposure would be ca 11,730 USD in Urban & 360 USD in Rural, i.e. a factor of 32 in per capita exposure between Urban & Rural. Both of these indicators are big and need to be corroborated by other socio-economic evidence given that most ETH houses are "mud & brick" type. The differences in urban and rural housing in Ethiopia need to be investigated to gain more insights. The 2007 census gives data on type of outer walls, roof cover, floor, ceiling but also other factors that influence the RV of a house. For the time being the only available resource is the 2007 Census, as the 2019 Census was indefinitely postponed.

**Comment [A20]:** Please add the statement mentioned in the Supplement i.e. "The inundation associated with each return period is assumed to occur everywhere simultaneously".

**Comment [A21]:** This would be expected to differ in urban and rural parts?

[Figure]

**3. Results and discussion**

**3.1. Urban-rural identification**

The results of our classification of ImageCat cells for Ethiopia into urban or rural are shown in Table 5, along with summaries of data from other data sources. For rural areas, our result (7.2%) is similar to that of GHS-SMOD (6.4%), which is the only other data source among the products that has a specific value for rural areas. The area in Ethiopia categorized as urban or built-up is relatively low in all data sources  and is in accordance with Ethiopia being one of the least urbanized countries in Sub Saharan Africa, with the share of urban population being only between 11% and 16% (Schmidt and Kedir, 2009).

> **Comment [A22]:** More recent datasets suggest: UN World Urban Prospects report (for 2014) Ethiopia Urban Popul. 19%, World Bank's 2016 estimate is at 19.9%.

[revised manuscript text omitted]

---

## Author Comment (AC1) · 20 Jun 2019

The comment was uploaded in the form of a supplement:
https://www.nat-hazards-earth-syst-sci-discuss.net/nhess-2019-32/nhess-2019-32-AC1-supplement.pdf

---

## Author Comment (AC2) · 20 Jun 2019

**Author Comment to RREFEREE 2**

*Interactive* comment on "Enhancement of large-scale flood damage assessments using building-material-based vulnerability curves for an object-based approach" by J. Englhardt et al.

5

[*RC2\_1*]: This is a very good paper, focusing on the importance of using building-material-based information in the exposure, vulnerability components of large-scale (global) flood modelling efforts.

[Our response]: We would like to thank the referee for the time put into the reviewing and the very valuable feedback that helped to improve the manuscript. We are pleased that the reviewer finds it a very good paper.

15 [RC2\_2]: The paper demonstrates clearly how such work is making significant improvements in flood risk assessment. Another important part is the discussion of spatial capture of urban-rural areas. This merits to also be included in the paper's title.

[Our response]: We thank the referee for this comment. We will follow the referee's suggestion to highlight the distinction of risk in urban and rural areas as an important part of our study and adjust the title to "Enhancement of large-scale flood risk assessments using building-material-based vulnerability curves for an object-based approach in urban and rural areas" (see here also our reply to comment RC1\_2 of referee 1).

[RC2\_3]: My review focused more on this aspect of the paper's content. Please see my comments in the attached PDF file.

25

[Our response]: We thank the referee for the feedback. All comments have been numbered and copied into this response document for ease of replying to them.

[*RC2\_4*]: I am concerned that the estimation of the replacement value of the buildings in Ethiopia shows a big urban-rural divide (buildings per capita exposure being 32 times greater in the urban areas vs the rural areas).

[Our response]: Please see our reply to comment RC2\_A19 of referee 2.

[RC2\_5]: Since this paper is applying the proposed methodology to Ethiopia it is very important to use Ethiopia data. It is necessary to revise the entire section "Object-based exposure data" to include review of the 2007 Ethiopia census.

- 5 [Our response]: The data for the last Ethiopian census were collected in May and November 2007 in both urban and rural areas and since then has seen considerable economic growth (World Bank, 2019a), but unfortunately the already delayed 2017 census was recently further postponed (Reuters, 2019). Two types of questionnaires were used in 2007, whereby a long questionnaire including housing characteristics was administered to 20% of randomly selected households (CSA, 2012). According to the census, the majority of all housing units in Ethiopia were of 'wood and mud' wall material (73.9%),
- 10 followed by 'wood and thatch / wood only' (13.0%), 'stone and mud' (7.1%), and only minor shares by several other wall materials. As pointed out by the reviewer in comment RC2\_A3, this amounted to about 80% of urban units assigned to the mud and wood type of wall materials compared to 72.5% in rural areas where also a large portion (15.5%) of units are of the wood and thatch / wood only type (CSA, 2010). It is part of the ImageCat methodology to apply census-based data which is redistributed and derived to a finer resolution given earth-observation (EO) indicators. EO is used to segment the region into
- 15 various development patterns which are used for stratified sampling of building characteristics. This approach provides both spatial focusing of assets beyond the census level, which is required for flood risk analysis, and a characterization of the spatial distribution of building characteristics beyond what is typically available in the data (Huyck and Eguchi, 2017). In all Ethiopian censuses, however, urban areas are defined as localities with 2,000 or more inhabitants, plus the capitals of all regions and sub-zones and further all localities with at least 1,000 people who are primarily engaged in non-agricultural
- 20 activities as well as other areas declared urban by administrative officials (Schmidt and Kedir, 2009). Therefore, also many smaller settlements are included as urban in the census and different definitions such as thresholds of built-up or population density, or a methodology using building stock like our approach can affect the urban-rural classifications and thus the distributions in these areas. Regarding the entire Ethiopian building stock, ImageCat estimates for the building structure types were initially developed through interviews with local professionals; and confirmed, cross-checked and adjusted with
- site surveys, scholarly journals (e.g. WHE), visual assessments/sampling process from satellite imagery. Information from the GEM Foundation were provided by the Earthquake Risk Consortium and were also used to "sanity check" the estimates. Obtaining the housing data can be more difficult than the population data and a consistent approach between countries was a goal of the ImageCat project. If we compare for example class I and II constructions in the ImageCat data (71.6% of the total building stock) to the 2007 census (approximately 97%), the differences are not surprising: Masonry construction is
- 30 minimal in the 2007 census (2.4%), and reinforced concrete seems non-existent (perhaps included in the "others" category, which accounts for 0.4% of the total building stock), but as observed in the ImageCat project such construction make up the majority in large cities. Furthermore, Ethiopia experienced "strong, broad-based-growth averaging 10.3% [GDP growth] a year from 2006/07 to 2016/17" which appears to coincide with a growth in the construction industry (World Bank, 2019b). For example, based on online imagery and ground observations in the ImageCat project, the sprawl observed through

historic satellite imagery since 2007 in Addis Ababa, appears to be a majority of class III and IV. We acknowledge the different results compared to the 2007 census data, and reasons for that discussed here, need to be better highlighted and we will include some information in the revised manuscript (p.22 l.27ff.) (please see also our reply to comment RC2\_6).

**5 p.22 l.27ff.**

25

"Nonetheless, as previously discussed in section 3.1, exposure of an area can vary depending on the applied dataset. Using ImageCat data, over half of the construction types in Ethiopia belong to class I, and about 14% towards each of the other classes (see Table 10). However, according to data from the last census in Ethiopia from 2007, 73.9% of all housing units in Ethiopia were of 'wood and mud' wall material, with 80% of the urban units and 72.5% of rural units, whereas a large share

- 10 of rural units were built with wood (and thatch) walls (15.5%). Compared to the ImageCat data, the Ethiopian building stock appears to be less diverse and shows a different distribution of urban and rural constructions, which is also affected by the applied definition of urban in the census. Since the 2007 census, Ethiopia has experienced considerable economic growth that appears to coincide with growth in the Ethiopian construction industry (World Bank, 2019). Furthermore, census data are aggregated to administrative levels and thus cannot be applied in the approach developed in this paper, for which an
- 15 object-based dataset is required that is also comparable between countries, such as the ImageCat data. With different methodologies in exposure datasets, future research should explore how flood risk assessments that are based on buildingmaterial-based vulnerability are affected by the applied building stock dataset and their different scales."

[RC2\_6]: Once this is done it will be also apparent that the section "3.2. Flood risk assessment" also needs to be revised
because the building stock distributions of classes I to IV in Ethiopia are quite different to what the authors have probably assumed. In this section a Table of classes I, II, III, III2 and IV distribution of the building footprints in urban and rural Ethiopia used in the model is not shown and this is an important omission.

This part of the work, i.e. the passing from census data to classification of the building vulnerability classes and the building footprints needs to be much more clearly explained than it is in the present version with some additional references for the ImageCat methodology.

[Our response]: Our study presents an approach for using building-material-based vulnerability in large-scale flood risk assessments. As described in the introduction chapter, traditional models aggregate the exposed elements into land-use categories, whereas in our alternative approach we are using the object-based data from ImageCat. As such, the Ethiopian

30 census data that the reviewer suggests cannot be directly applied and has several disadvantages. For example, compared to the ImageCat data, the census data are aggregated to administrative levels that have different spatial extents and are not comparable throughout the country. In our flood risk assessment, we can overlay the inundation maps with the finer resolution dataset from ImageCat to identify exposed areas. Moreover, the Ethiopian census follows a methodology set out by the country's statistical agency, meaning that the definitions of urban and rural areas are different to those in other

countries, which is contradicting to one of the aims of this study to develop a methodology that could also be used in other regions. Furthermore, using census data for a building-material-based approach would require going back to a model setup up similar to land-use-based flood risk models due to the aggregation in the census data. In our manuscript Ethiopia is an example to which we apply the approach we developed. Using large-scale datasets that have a consistent methodology to

- 5 provide exposure data for many countries such as the object-based ImageCat data allows us to analyse flood risk based on building material vulnerability outside of resource-intensive local studies and apply one approach in order to achieve comparability between countries. In combination with the adjustments to the manuscript in response to comment RC2\_5, more information will be included on the differences between datasets and an overview of the building stock distribution in the ImageCat data (p. 22 1.27ff. and Table 10). Finally, we like to point out that we are currently working on a follow-up
- 10 paper which focuses on analysing different approaches and compares flood risk assessments for several countries using different building exposure datasets. Regarding the building footprints, Table 10 in combination with the overview in Table 3 allows the reader the reproduction of building footprints per class and land use. We will also add some more information regarding the ImageCat estimation of building area to the manuscript. (See here also comment RC2\_A1).

**15 p.23 l.16**

**"Table 101 Ethiopian building stock according to ImageCat data"**

| Туре | Description                                        | % total
building
stock | Class | % urban
building
stock | % rural
building
stock |
|------|----------------------------------------------------|------------------------------|-------|------------------------------|------------------------------|
| ADB  | URM adobe building                                 | 4.1                          |       |                              |                              |
| ERTH | Earthen building                                   | 3.9                          | т     | 2.4                          | 72.0                         |
| INF  | Informal building                                  | 9.4                          | 1     | 3.4                          | 72.0                         |
| WWD  | Wattle & daub building                             | 39.7                         |       |                              |                              |
| WLI  | Light wood building                                | 1.0                          | п     | 2.0                          | 10.0                         |
| WS   | Solid wood building                                | 13.5                         | 11    | 2.0                          | 18.0                         |
| BRK  | URM brick building                                 | 6.1                          |       | 20.0                         | 10.0                         |
| STN  | URM stone building                                 | 8.2                          | 111   | 29.9                         | 10.0                         |
| RC   | Reinforced concrete frame with URM infill building | 13.9                         | IV    | 64.8                         | 0.03                         |

See RC2 supplement https://www.nat-hazards-earth-syst-sci-discuss.net/nhess-2019-32/nhess-2019-32-RC2-supplement.pdf [RC2\_A1]: Census data usually report the number of housing units (incl. in Ethiopia). Some explanation as to how these data have been used to derive information on the number of residential and non-residential "buildings" is needed.

[Our response]: We thank the reviewer for the comment. Given that most of the residential building stock is single family

5 housing, the number of housing units is used directly from the census data in the ImageCat data and in there, apart from the development patterns, not further differentiated. We will include this in combination with further information on the ImageCat methodology (p.9 1.5ff.). (See here also comments RC2\_6).

p.9 1.5ff.

30

- 10 "For the building numbers the Ethiopian census data on housing units was used directly in most regions as the building stock is mostly single family housing. The living area was gleaned from sampling building footprint data, and as with structural characteristics, varies by development pattern. For a predominantly commercial pattern, building stock data is adjusted with exposure derived from building footprint data. The number of floors can vary by development pattern, but for the vast number of buildings is single story for most of the country. For highly urbanized areas the number of stories was adjusted through expert opinion of several country-based structural engineers (Huyck and Eguchi, 2017)."
  - [RC2 A2]: This Reference is missing

[Our response]: We apologize for the oversight and will include the reference.

20 [RC2\_A3]: In the 2007 census of Ethiopia the most common wall-type is "Mud and Wood" forming 80% of houses in Urban & 72.5% in Rural. In rural areas the next most common are "Wood and Thatch / Wood only" (15.5%).

[Our response]: Please see the response to comment RC2\_5.

25 [RC2\_A4]: In future, if the data allow, differentiating vulnerability between clay bricks, stones and concrete blocks should be considered.

[Our response]: We agree with the reviewer that future research would benefit from further differentiation within the current vulnerability classes, if and when sufficient data becomes available. We will add this suggestion to the manuscript in combination with our response to comment RC2 A5.

[RC2\_A5]: These buildings tend to have more non-structural elements that can be vulnerable to flooding especially in Africa's Urban areas e.g. air conditioning units, partition walls, mechanical & electrical components, etc. that would need to be considered both in terms of their contribution to the overall building replacement value and their vulnerability. At a future stage.

**5**

[Our response]: While the focus of our study is the structural vulnerability, we agree that future flood risk assessments would benefit from including further components of the buildings and will add this to the recommendations for future research (p.24. 1.28ff.).

10 p.24 l.28ff.

"Furthermore, if the data allows in the future, vulnerabilities within the classes could be further refined such as between clay, stone and concrete brick/block construction or regarding non-structural elements like electrical components and partition walls."

15 [RC2\_A6]: This needs a Reference and brief explanation of how it was developed. In particular how the number of floors was estimated.

[Our response]: Please see our response to comment RC2\_A1.

[*RC2\_A7*]: As use is also made of PAGE v2.0 classification system, an additional column is needed to map the ImageCat classes to the PAGER classes.

[Our response]: We thank the reviewer for this comment, and we will add an overview of assigned classes to the PAGER typology and include further information on the different construction types to the revised supplements (see supplementary section 1 and supplementary table 1) and add a note to that at table 2.

25

 $[RC2\_A8]$ : URM = unreinforced masonry, RC = reinforced concrete - must be added for the benefit of those less familiar with these acronyms

[Our response]: Please see our response to comment RC2\_A7.

30 [RC2\_A9]: DS & STN are similar. Briefly explain their differences.

[Our response]: Please see our response to comment RC2\_A7.

[RC2\_A10]: RC and C3 are similar Briefly explain their differences.

5 [Our response]: Please see our response to comment RC2\_A7.

[RC2\_A11]: "Earthen", "Mud walls", "Rammed earth", "Adobe" are very similar typologies. Briefly explain their differences.

10 [Our response]: Please see our response to comment RC2\_A7.

[RC2\_A12]: URM stands for "unreinforced masonry". BRK, CB, are very similar UFB, UCB respectively. Briefly explain their differences.

15 [Our response]: Please see our response to comment RC2\_A7.

[RC2\_A13]: Add also this Ref: Jaiswal, K. S., Wald, D. J., and Porter, K. A. (2010a). A Global Building Inventory for Earthquake Loss Estimation and Risk Management. Earthquake Spectra

20 [Our response]: We will include the suggested reference.

[RC2\_A14]: In PAGER for Africa there is the problem that only 19 of the 56 countries have original data (and of these 6 are from 1993), the rest are based on "neighbor country". Also these data are primarily distributions of the housing units, not the Residential + Non-Residential buildings, and in urban areas the building distributions would be quite different due to

25 many houses being in apartment buildings. Also differentiation for Urban-Rural and Residential-Non-Residential exists only for 2 countries (Algeria & Morocco). The value of 2% in Urban is for Algeria. In rural Algeria this value is 15%. In the 2007 Ethiopia Housing census the ratio of class I & II in Urban is 89% (81% in Addis Ababa) which would challenge the rural hypothesis (>50%)..Since this paper is examining Ethiopia it would be better to use the data from the 2007 Ethiopia Census (also available in PAGER v2) that gives distributions of the housing units in urban and rural areas or use the

30 distributions of Ethiopia in PAGER (though they do not differentiate urban-rural).

[Our response]: We thank the reviewer for the comment and further clarified PAGER and the selection in the manuscript (p.10 1.4ff.). The literature provides only little information on differences between building stock in urban and rural areas, usually the focus is on one of the areas and/or housing durables and quality. However, the PAGER dataset provides estimates of building stock inventory on a global scale. The information basis for these estimates is better for some countries

- 5 than others and for many African countries the estimates are based on neighbouring countries. Therefore, we included in Figure 3 not only the distribution of class I and II construction types in urban and rural areas for Africa, but also for different income groups. The average class I and II share in urban areas is higher (10%) for the low and lower middle income countries than the African average (2%), however there is a clear difference to rural areas with (36% class I and II in lower and lower middle income countries and 22% for African countries). This information in PAGER indicates that there are
- 10 distinct differences between the built environment in urban and rural areas. The threshold we set in our approach is set even higher (>50% class I and II), which means that an area classified as rural is dominated by more traditional and less expensive housing. We acknowledge in the manuscript that the presented approach to differentiate urban and rural can be applied if the building stock is more heterogeneous, but similarly to other products additional indicators for example population density could be further incorporated to refine the approach (p.18 1.18ff., p.24 1.21ff.). Furthermore, as we showed in section 3.1, the
- 15 urban-rural map derived with our approach is comparable to other maps that are classified from remote sensing data and/or using several input parameters.

**p.10 l.4ff.**

"To check the assumption that the share of class I and II buildings in developing countries is higher in rural areas compared to urban areas, we examined these shares in the PAGER dataset (Jaiswal and Wald, 2008; Jaiswal et al., 2010). PAGER is a global residential and non-residential building inventory at the country level (usually but not exclusively expressed in proportions of people living or working in particular building structure typologies in urban and rural areas respectively), which is often used in earthquake research. PAGER provides information at a country level on the construction types that make up the total urban and rural building stock., though the information quality is varying between countries. First, we

- 25 reclassified the PAGER construction types into the four flood vulnerability classes used in our study (see Supplementary table 1). Then, we calculated the percentage of buildings in PAGER's total urban and rural building stocks that are categorised as class I and II (Figure 3). The building stock differences between urban and rural areas can be found to a changing degree in all groups. While there is a distinct gap suggested for Africa, PAGER has to rely there on very limited information (i.e. only 2 of the countries differentiate urban and rural building stock without judging on information from
- 30 neighbouring countries). Nevertheless, the data for urban and rural building stock distribution compared by income level also indicates this differences in the built environment. In low and lower middle income countries, the percentage of buildings in class I and II is indeed much higher in rural areas (36%) than in urban areas (10%). These differences are far less pronounced for higher income countries. The chosen threshold to identify rural areas in the ImageCat dataset (>50%) is larger than the

average share we find in PAGER (Figure 3). This means that cells identified as rural using the ImageCat data information about the built environment with the chosen threshold are quite likely to indeed be rural."

[RC2\_A15]: Add in the Supplementary References: Congalon, 1991

5

10

[Our response]: We apologize for the oversight and will include the reference.

[RC2\_A16]: This reference is a GFDRR report, but "Replacement Cost Refinements to the Exposure data" is not included. As it is a crucial reference, it would be good to include a Reference where this would be explained. The same stands for the reference ImageCat et al. (2017), "Exposure Development for 5 Sub-Saharan African countries"

[Our response]: ImageCat et al. (2017) and Huyck and Eguchi (2017) are both included in the reference list. The Huyck and Eguchi (2017) reference is a report not yet published by GFDRR, about the ImageCat data for several African countries. This report also covers the ImageCat approach to estimate replacement costs. Further information then provided in these references or given in ecoephony in generation of the approach to estimate replacement costs.

15 references or given in accompanying articles (see for example references used on p.3 l.26ff.) is proprietary information of ImageCat.

[RC2\_A17]: Please provide more explanation as to why "Class II 2 floors" has nearly 5.6 times greater footprint than "Class II 1 floor".

**20**

25

30

[Our response]: We know from the ImageCat data that most of the buildings in these classes are larger, which is further confirmed by the ImageCat description for Ethiopia of the typical building stock in different areas which reports that those buildings are predominantly found in urban environments with for example many apartment blocks instead of single family buildings. We will adjust the building footprint description in the manuscript to reflect the difference and its explanation (p.13 1.13ff.).

**p.13 l.13ff.**

"The buildings of class III and IV with multiple floors have a much greater footprint than the one assigned to the other classes. While buildings with smaller footprints are primarily single family residential structures or within informal settlements, the buildings of the last two classes are mainly found in urban environments, with many of them being long apartment blocks with very large building footprints leading to a larger average footprint. The resulting building footprints for Ethiopia can be seen in Table 3."

[Our response]: Please see our response to comment RC2\_A17.

5

[RC2\_A19]: These values when summed would suggest that the replacement value of Ethiopia's building stock is assessed at 384% of 2016 GDP. The per capita buildings exposure would be ca 11,730USD in Urban & 360USD in Rural, i.e. a factor of 32 in per capita exposure between Urban & Rural. Both of these indicators are big and need to be corroborated by other socio-economic evidence given that most ETH houses are "mud & brick" type. The differences in urban and rural housing

10 in Ethiopia need to be investigated to gain more insights. The 2007 census gives data on type of outer walls, roof cover, floor, ceiling but also other factors that influence the RV of a house. For the time being the only available resource is the 2007 Census, as the 2019 Census was indefinitely postponed.

[Our response]: We thank the reviewer for this comment. In this paper we present a large-scale flood risk assessment approach that is particularly interesting for areas where there is a large variation in construction types, and provide the application in Ethiopia as an example. Therefore, when calculating the maximum damage values, we are using the Huizinga et al. (2017) dataset of country-specific construction costs based on a globally consistent process for a non-biased comparison between different countries and differentiate from it maximum damage values for our vulnerability classes. Huizinga et al. (2017) describe in their report that information available about flood damage and construction cost values in Africa to inform their approach is very sparse. Consequently, for many countries, especially low income countries, it is more

- difficult to reproduce the construction costs and they applied non-linear regression for better representation. We acknowledge that the difference in the total values we calculate for urban and rural areas is high. Firstly, this can in part be attributed to the fact that urban areas are defined in our approach by a greater proportion of higher value buildings (class III and IV). As we discuss in the manuscript (p.18 1.20ff., p.24 1.22ff.), this can lead to a higher exposed value of the urban
- 25 built environment, as for example urban slums could be misclassified as rural areas. As pointed out by the reviewer, the Ethiopian building stock value in our study surpasses its 2016 GDP. However, a country's building stock is created over decades and continuously developed and can therefore exceed GDP. For example, when taking the 2016 GDP and value of all dwellings from the Dutch statistical office, even in the Netherlands the residential building stock has a value of 245% of the country's GDP and the per capita exposure is 102,000USD (CBS, 2019). We might also look at GDP exceeding damage
- 30 and losses from natural disasters, according to IWF studies for example events on the pacific islands such as cyclone Nigel in Vanuatu with damage of 131% of the country's GDP in 1985 or in the Caribbean like the 2010 Haiti earthquake with about 120% (Cabezon et al., 2015; Lee et al., 2018). Secondly, while there is no data that would be sufficient to quantify gaps in urban and rural exposure, some information can indicate the level of difference in urban and rural areas. According

to the census data the rural population in Ethiopia is 5 times the urban population, and out of the 90% of the rural housing units that own livestock, in more than half of them the livestock spend the nights in a room with people. Furthermore, considering household size the difference between urban and rural is already reduced to factor 25 as more people live in rural households, with an average rural household size at 4.9 persons (urban 3.8) (CSA, 2010). Literature about indicators that

- 5 might inform differentiations between urban and rural housing are mostly surveyed for households in urban areas (e.g. Adeoye, 2016; Gulyani et al., 2018), or regarding low-cost and informal living in urban areas (e.g. Govender et al., 2011; Simiyu et al., 2018), and/or are focused on the living conditions in terms of health and sanitation (e.g. Ashebir et al., 2013; Sahiledengle et al., 2018). The Demographic and Health Survey for Ethiopia also showed large differences for flooring material in urban and rural households which was the only structural characteristic surveyed: While about 67% of urban
- 10 households have higher quality floors1, only about 4% of rural floors are of these types (CSA and ICF, 2016). Also the 2007 census shows that over 86% of urban housing units get their drinking water from taps in- or outside the house or compound compared to only 15% for rural ones which otherwise use wells, springs, river, etc. as their source; similarly, 75% of rural housing units have no toilet facility which is the case for 28% in urban settings (CSA, 2010). Such differences in drinking water, sanitation and floor material illustrate that there are large differences for the living conditions in the two areas and
- 15 give an indication about the difference in exposed value. In order to better illustrate the urban and rural gap, we will include information about housing quality to the end of section 2.3 (p.14 1.10ff.).

p.14 l. 10ff.

- 20 "Similarly, there is also a large gap between the living standard in rural and urban areas. The last Ethiopian census in 2007 (CSA, 2010) and the 2016 DHS report (CSA and ICF, 2016) provide some indications for rural and urban households that show huge differences in household durables and quality, for example more than half of the rural household with livestock share at night the room with the animals, or high quality floors in two thirds of urban households compared to only 4% of floors in rural households. The contrasts shown there in housing characteristics such as sanitation, drinking water and
- 25 flooring material illustrate that there are large differences in living conditions which indicate similar differences in exposed urban and rural value."

[RC2\_A20]: Please add the statement mentioned in the Supplement i.e. "The inundation associated with each return period is assumed to occur everywhere simultaneously".

30

[Our response]: We will include the statement in the revised manuscript (p.14 1.18).

<sup>1 Parquet or polished wood, vinyl or asphalt strips, ceramic tiles, cement, carpet

[Our response]: While urban areas often seem to have better flood protection than rural areas, Scussolini et al. (2016) do not differentiate their data and no further information on protection standards is available.

5

[RC2\_A22]: More recent datasets suggest: UN World Urban Prospects report (for 2014) Ethiopia Urban Popul. 19%, World Bank's 2016 estimate is at 19.9%.

[Our response]: We will adjust the statement to include more recent urban population estimates (p.151.19).

**10**

**p.15 l.16ff**

"The area in Ethiopia categorized as urban or built-up is relatively low in all data sources and is in accordance with Ethiopia being one of the least urbanized countries in Sub Saharan Africa, with the share of urban population being according to Schmidt and Kedir (2009) only between 11% and 16%, or according to more recent data from the World Bank (2016) at show 20%

15 about 20%."

[RC2\_A23]: This may not be the case in Ethiopia as suggested by the 2007 housing census

[Our response]: Please see our response to comment RC2\_5, RC2\_6 and RC2\_A14.

**20**

[Our response to referee's grammatical/typo corrections and rephrasing]: We like to thank the referee for pointing out parts of the text that needed corrections or where additional information was suggested to provide for further clarification for the reader. The manuscript was adjusted where necessary.

[revised manuscript text omitted]

material                           | Event / applied area                                                                             |
|----------------|----------------|---------------------------------|-----------------------------------------------------------------------------|------------------------------------------------------------------------------|--------------------------------------------------------------------------------------------------|
| Ι              | India          | Dhillon (2008)                  | Field survey                                                                | Mud struct ur s <del>.</del>                                          | Birupa River basin in Orissa after the 2006 flood                                                |
| Ι              | India          | Maiti (2007)                    | Household interviews                                                        | Mud wall b ui ld in gs <del>.</del>                            | Rural areas in Orissia after the 2003 flood                                                      |
| Ι              | China          | Li et al. (2016)                | Interviews, questionnaires, field investigation                             | Wood-earth struct ure s-                                              | Taining county town, Fujian province                                                             |
| Ι              | Malawi         | Rudari et al. (2016)            | To generic Malawi housing
typology adjusted CAPRA                        | Traditional (mud walls), semi-
permanent (sun-dried bricks)
typologies | Based on data for Northern and Central Malawi                                                    |
| II             | India          | Dhillon (2008)                  | Field survey                                                                | Wooden struct ure s <del>.</del>                                      | Birupa River basin in Orissa after the 2006 flood                                                |
| II             | Germany        | Buck (2007)                     | Expert seminar                                                              | Wood struct ure s <del>.</del>                                        | Bldgs. in flood prone areas of Greifswald                                                        |
| Π              | New

[revised manuscript text omitted]